# Variable microtubule architecture in the malaria parasite

Josie L. Ferreira[1,2,3,10,11], Vojtěch Pražák[1,2,4,11], Daven Vasishtan[1,2,4], Marc Siggel [1,5], Franziska Hentzschel[6,7], Annika M. Binder [6], Emma Pietsch[1,3,8], Jan Kosinski [1,5,9], Friedrich Frischknecht[6,7], Tim W. Gilberger[1,3,8] & Kay Grünewald [1,2,4,8] ✉

Microtubules are a ubiquitous eukaryotic cytoskeletal element typically consisting of 13 protofilaments arranged in a hollow cylinder. This arrangement is considered the canonical form and is adopted by most organisms, with rare exceptions. Here, we use in situ electron cryo-tomography and subvolume averaging to analyse the changing microtubule cytoskeleton of *Plasmodium falciparum*, the causative agent of malaria, throughout its life cycle. Unexpectedly, different parasite forms have distinct microtubule structures coordinated by unique organising centres. In merozoites, the most widely studied form, we observe canonical microtubules. In migrating mosquito forms, the 13 protofilament structure is further reinforced by interrupted luminal helices. Surprisingly, gametocytes contain a wide distribution of microtubule structures ranging from 13 to 18 protofilaments, doublets and triplets. Such a diversity of microtubule structures has not been observed in any other organism to date and is likely evidence of a distinct role in each life cycle form. This data provides a unique view into an unusual microtubule cytoskeleton of a relevant human pathogen.

Microtubules are a vital cytoskeletal component across all branches of eukaryotes, forming tracks for intracellular transport, structural support for locomotion and spindles for chromosome segregation. The polymerisation of α-/β- tubulin heterodimers forms protofilaments which interact laterally to assemble into a hollow cylinder. The 13-protofilament microtubule is considered canonical as this structure has been most commonly observed within widely studied eukaryotic supergroups. While early electron microscopists found multiple examples of non-canonical microtubules[1], the difficulty in purifying functional native tubulin from some species, and the overreliance on model organisms, resulted in our understanding of microtubule biology being based primarily on the study of metazoan microtubules. Non-canonical examples, such as the 11-protofilament microtubules of specialised cells in nematodes[2,3] and the 15-protofilament microtubules of inner-ear pillar cells[4,5], are considered to be curious outliers. The result is that, although protozoa are a large and diverse group, much of our hypotheses about their microtubules have been inferred from metazoan studies.

The eukaryotic protozoan parasites of *Plasmodium* spp., rely on a scaffold of ordered subpellicular microtubules (SPMTs) as their main structural cytoskeletal components. *P. falciparum*, the causative agent of malaria, has a complex life cycle alternating between mosquito

[1]Centre for Structural Systems Biology, Hamburg, Germany. [2]Leibniz Institute for Virology (LIV), Hamburg, Germany. [3]Bernhard Nocht Institute for Tropical Medicine, Hamburg, Germany. [4]Division of Structural Biology, Wellcome Centre for Human Genetics, University of Oxford, Oxford, UK. [5]European Molecular Biology Laboratory, Hamburg, Germany. [6]Integrative Parasitology, Centre for Infectious Diseases, Heidelberg University Medical School, Heidelberg, Germany. [7]German Center for Infection Research, DZIF Partner Site Heidelberg, Heidelberg, Germany. [8]University of Hamburg, Hamburg, Germany. [9]Structural and Computational Biology Unit, EMBL, Heidelberg, Germany. [10]Present address: Institute of Structural and Molecular Biology, Birkbeck, University of London, London, UK. [11]These authors contributed equally: Josie L. Ferreira, Vojtěch Pražák. ✉e-mail: kay.gruenewald@cssb-hamburg.de

vector and human host (Fig. 1). Extensive coevolution with the two hosts has resulted in the pathogen utilising a multitude of highly specialised and morphologically distinct cell types, here referred to as forms. Although each form is a master of a distinct niche and morphologically varied, the presence of SPMTs is a unifying feature. SPMTs lie below and interact with a double membrane known as the inner membrane complex (IMC) located beneath the plasma membrane. Together these structures are referred to as the pellicle which is found across Apicomplexa, including, for example, *Toxoplasma gondii*. The pellicle is broken down in transition and then rebuilt during the maturation of each life cycle form, and each time it is accompanied by a different set of associated proteins[6,7]. This de novo re-organisation is a driving force behind the parasite's substantial morphological changes and SPMTs play a key role in this process[8].

Although eukaryotic tubulins are highly conserved, there are appreciable differences which make apicomplexan microtubules stand out. The parasite's SPMTs are extremely stable, resistant to most classical depolymerisation protocols such as cold treatments[9], addition of microtubule depolymerising drugs[10] and detergents[11,12]. Another feature unique to apicomplexan tubulin is its formation of L-shaped semi-tubules which, with accessory proteins, form a helical structure termed the conoid[13,14]. The conoid is a specialised structure involved in invasion. Although characterised in *Toxoplasma*, homologues of some conoid components are expressed in *Plasmodium* spp.[15].

Microtubules assembled in vitro, from purified tubulin, consist of 9 to 16 protofilaments, with different distributions of protofilament numbers depending on the species and conditions used[1,16]. As these variations are not normally found in cells, stipulation of a uniformly 13-protofilament microtubule population therefore needs to be established during nucleation by external factors. There are four common mechanisms for controlling the number of protofilaments in a microtubule: templating via the γ-tubulin ring complex (γTuRC)[17,18], setting a specific inter-protofilament angle through lateral binding of proteins such as doublecortin[19,20], expression of different tubulin isoforms[21] or post-translational modifications[22]. In metazoans, nucleation of microtubules usually occurs at centrosomes (microtubule organising centres, MTOCs). In the invasive forms of apicomplexan parasites, a structurally diverse MTOC located at their apical end, the Apical Polar Ring (APR), coordinates the higher order spatial control of SPMTs[23], but the specific mechanism remains unknown.

To directly visualise the diversity of microtubules in *Plasmodium* spp, we image four different parasite life cycle forms under native conditions using cryogenic focussed ion beam (FIB)-milling, electron tomography (cryo-ET) and subvolume averaging (SVA) (Fig. 1). By revealing these structures in their native context, in 3D, we uncover an unusual cytoskeletal architecture, where each parasite form has a distinct, specialised microtubule structure, some being substantially different from the canonical microtubule.

## Results

### Visualisation of native microtubules throughout the *Plasmodium* life cycle by in situ electron cryo-tomography

To determine the in situ structures of microtubules in different *Plasmodium* forms, we performed FIB-milling and cryo-ET on two mosquito (sporozoites and ookinetes) and two human forms (merozoites and gametocytes, Fig. 1). We performed targeted FIB milling (Fig. 1b) to produce lamella with thicknesses between 50 nm and 300 nm at locations containing parasites. Initially, correlative light and electron microscopy (CLEM) was used to identify parasites vitrified on EM grids. We later found that their distinctive shapes in SEM (Fig. 1c, column 2) allowed targeting without correlation. For each stage we acquired between 50 and 100 tomograms from ~20 lamellae, to obtain good coverage of different subcellular regions. We performed manual picking in order to perform SVA exhaustively on all microtubules, thus

generating complete ultrastructural models and allowing statistical analysis. To avoid potential bias, all ~850 individual microtubules in the dataset were analysed independently with SVA. This allowed us to investigate structural variability, including the number of protofilaments and relative polarity[24], both within each cell and between the four life cycle forms.

### Sporozoite and ookinete SPMTs are canonical microtubules with a twist: an interrupted luminal helix

We started our study by imaging parasite forms isolated from the mosquito vector: sporozoites and ookinetes. These motile forms are elongated with a dense SPMT scaffold underneath the IMC. At first glance, the lumen of SPMTs in both sporozoites and ookinetes contained a (pseudo-) helical density with ~8 nm periodicity, consistent with previous predictions (Figs. 2, 3)[11]. To generate an unbiased EM density maps, 422 sporozoite and 177 ookinete SPMTs were analysed by SVA without applying any (pseudo-)symmetry. Both of the resulting structures showed 13-protofilament microtubules with twice-interrupted luminal helices (Figs. 2b–d, 3c). Similar structures, named Interrupted Luminal Helices (ILH), were first observed in flagellar ends of human spermatozoa[25], and more recently in equine and porcine spermatozoa[26], and tachyzoites of *Toxoplasma gondii*[14,27]. We adopted the nomenclature of Zabeo et al[25]. The *T. gondii* ILH consists of thioredoxin-like proteins 1 and 2 (TrxL1, TrxL2) and subpellicular microtubule protein 1 (SPM1). We hypothesised that the ILH in *Plasmodium* and *Toxoplasma* likely consist of homologous proteins: firstly, *Plasmodium* spp. have homologues of TrxL1 (PfTrxL1) and SPM1 (PfSPM1) but lack TrxL2, which in the *Toxoplasma* ILH is present in both, and completely fills one of the two interruptions. Consistently, we exclusively observed twice-interrupted luminal helices in *Plasmodium*. Secondly, compatible with the ILH being composed of SPM1 and TrxL1 in *Plasmodium*, we saw high expression in *Plasmodium berghei* (Pb) of PbSPM1-GFP and PbTrxL1-GFP in our endogenously-tagged sporozoite lines (Fig. S2). Finally, *Plasmodium* and *Toxoplasma* TrxL1 and SPM1 are highly conserved[12] (Fig. S1a, b), and the model of *T. gondii* SPMT assembly (pdb 7MIZ) fits well into our EM maps (Fig. 2c, d). Due to the asymmetric nature of the ILH, there is only one way that the ILH structure from *T. gondii* can be fitted into the EM density of *Plasmodium* SPMTs. We therefore suggest that *P. falciparum* ILH consists of 10 copies of PfTrxL1, likely with an equivalent number of PfSPM1 separated into two half-crescents (Fig. 2c). This also provided us with a reliable method of identifying both the seam position and alpha-beta tubulin location in our structure (Fig. 2c, d).

Although most microtubule structures solved to date are close to circular in cross section, *Plasmodium* microtubules with ILH and to a lesser extent *T. gondii* microtubules with ILH (Fig. S1c, d) are flattened along an imaginary axis roughly between protofilaments 2 and 9. This implies that the ILH stabilises the inter-protofilament angle at ~26°, deviating by ~2° from the theoretical 27.7° of canonical 13-protofilament microtubules (Fig. S1c, d). The deviation from circular cross-section accumulates over 5 subunits and is then compensated for by 37° relative angle between protofilaments 6 and 7, and 12 and 13. Comparisons of the predicted structure of *Plasmodium's* TrxL1 to that of *T. gondii* showed that the largest difference is at the N-terminal helix (Fig. S1e), which is responsible for a large part of the subunit-subunit interface and may play a role in the increased ellipticity.

### Ookinetes contain a conoid

At the apical end of the ookinete, we observed a structure consistent with a classical conoid made up of a unique tubulin structure (N = 1, Fig. 3e). The average volume cross section is consistent with the L-shaped tubulin-based conoid described in *T. gondii*[14]. The ookinete conoid measured 70 nm in height and 300 nm in diameter, resembling a retracted state. Until recently, *Plasmodium* (belonging to the class *Aconoidasida*, meaning conoid-less) was thought to not possess a

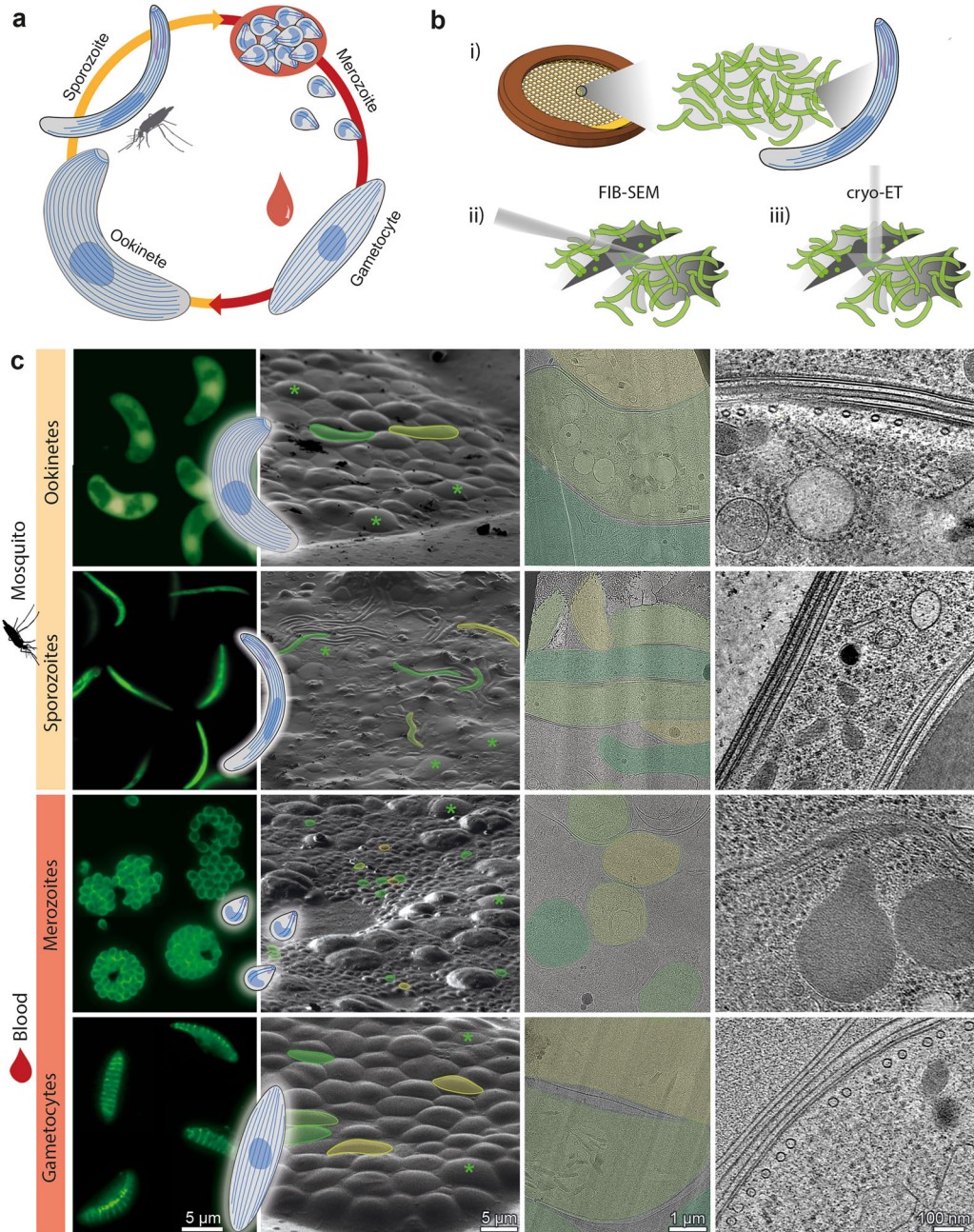

**Fig. 1 | Imaging parasites across the *Plasmodium* life cycle: from live parasites to high-resolution 3D volumes. a** Simplified *Plasmodium* life cycle of parasite forms studied here. Sporozoites are injected into the host. After differentiating in the liver, merozoites are released into the blood. The majority enter an asexual replication cycle (merozoites) and a small percentage commit to becoming gametocytes. Gametocytes are taken up by a mosquito and after fusion of male and female gametes in the mosquito gut, zygotes transform into ookinetes. Ookinetes cross the mosquito midgut, develop into oocysts and form thousands of sporozoites, which migrate to the salivary glands. **b** Schematic representation of our workflow: (i) live parasites are vitrified on EM grids. (ii) cells are thinned into lamella and then (iii) imaged by electron cryo-tomography (cryo-ET). Tilt-series are collected and computationally reconstructed into 3D volumes. **c** Columns 1–4: representative images of parasites at different workflow steps. 1: compositions of fluorescence images of cells highlighting overall parasite shape. Inset: cartoon representation of each stage. 2: Scanning Electron Microscopy (SEM) micrographs showing *Plasmodium* parasites (some false-coloured in yellow and green) surrounded by host cells (green asterisks). 3: micrographs showing overviews of lamellae. 4: slices through example tomograms.

conoid. Our data validate previous genetic, fluorescence microscopy, and classical EM data hinting at the presence of a conoid-like structure at the ookinete apex[15,28].

## Merozoite microtubules are canonical and lack an interrupted luminal helix

Based on our mosquito form data and the recent structures from *T. gondii*, it seemed plausible that the ILH is an ubiquitous feature of apicomplexan microtubules. To verify that this is a universal feature in all life cycle forms, we next analysed *P. falciparum* forms from the human host. Furthermore, to assess whether the presence of an ILH is specific to SPMTs or a general microtubule component, we imaged at two timepoints of schizont development. For SPMTs, we imaged fully segmented schizonts (merozoites), while for spindle microtubules we aimed for dividing schizonts. The asexual blood stage, the merozoite, which egresses from a mature schizont, is a small cell with only two to three SPMTs (Fig. 4). Merozoites are short-lived; to avoid imaging non-viable cells, we stalled schizonts prior to egress using a well-

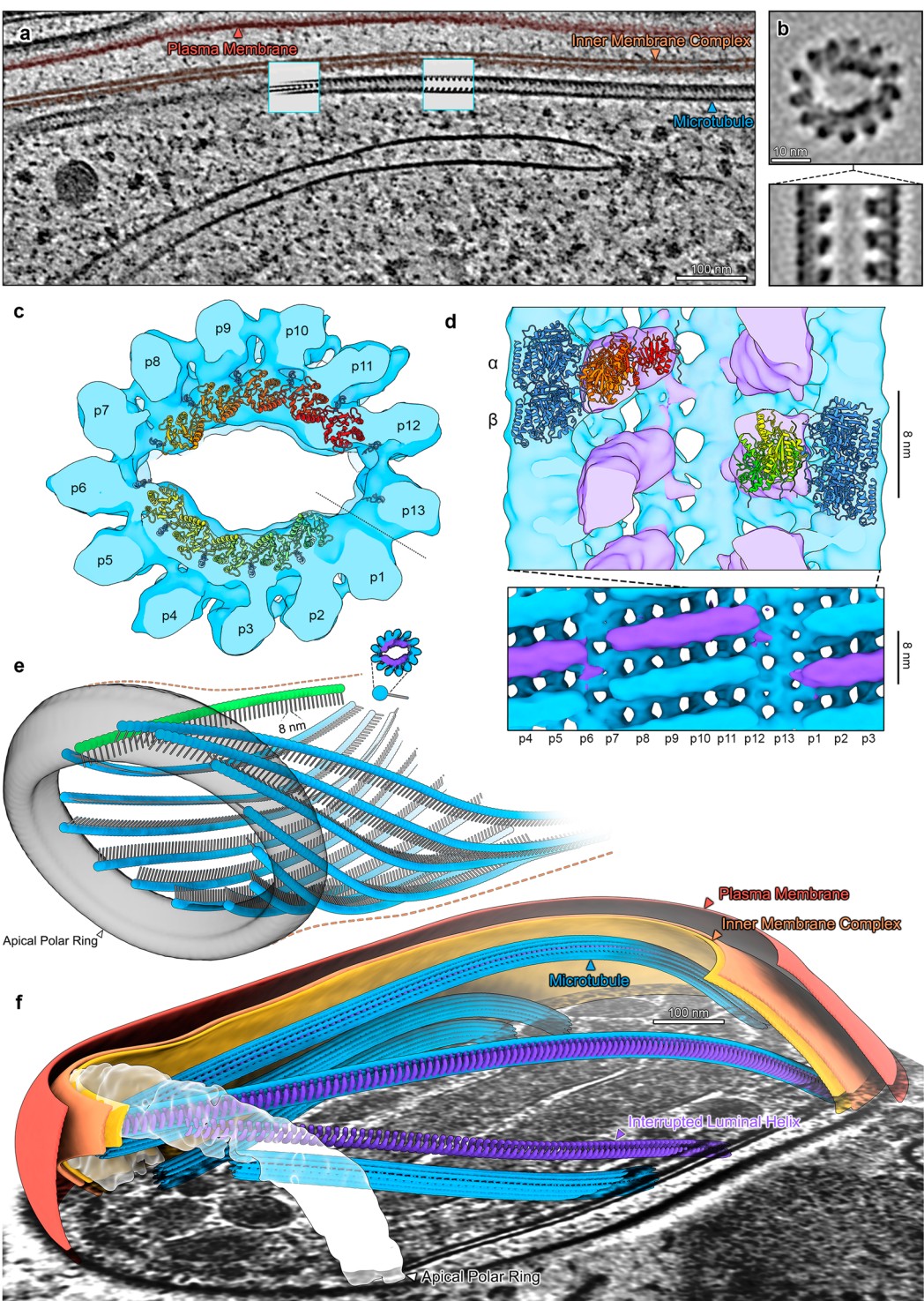

**Fig. 2 | Sporozoite subpellicular microtubules (SPMTs) contain a periodic luminal density inside 13 protofilament microtubules. a** Slice through a tomogram illustrating the overall architecture of the pellicle within a cell. One SPMT is seen weaving in and out of the slicing plane. Insets: slices through EM map (shown in B, C, D) placed back into the tomogram at positions determined by SVA. **b** Orthogonal slices through the EM map. **c** Isosurface representation of the EM map with pseudoatomic model (7MIZ) fitted into the EM density. p1-p13 = protofilament numbers, dotted line = seam position. **d** Top: section through the EM map showing the position of an α/β tubulin dimer relative to TrxL1. Bottom: radial projection with one period of the ILH highlighted in purple. **e** The apical pole of a *P. falciparum* sporozoite with a full set of microtubules (13 blue + 1 green). The APR is represented by an isosurface, the SPMTs as pin models, where the pinhead marks the centre and the line is oriented towards the seam. **f** Segmented sporozoite apical pole. The tubulin density (13 protofilaments) of two SPMTs was hidden to reveal the ILH. Unannotated slice through tomogram shown in Fig. S3a and the full volume in supplemental movie S1.

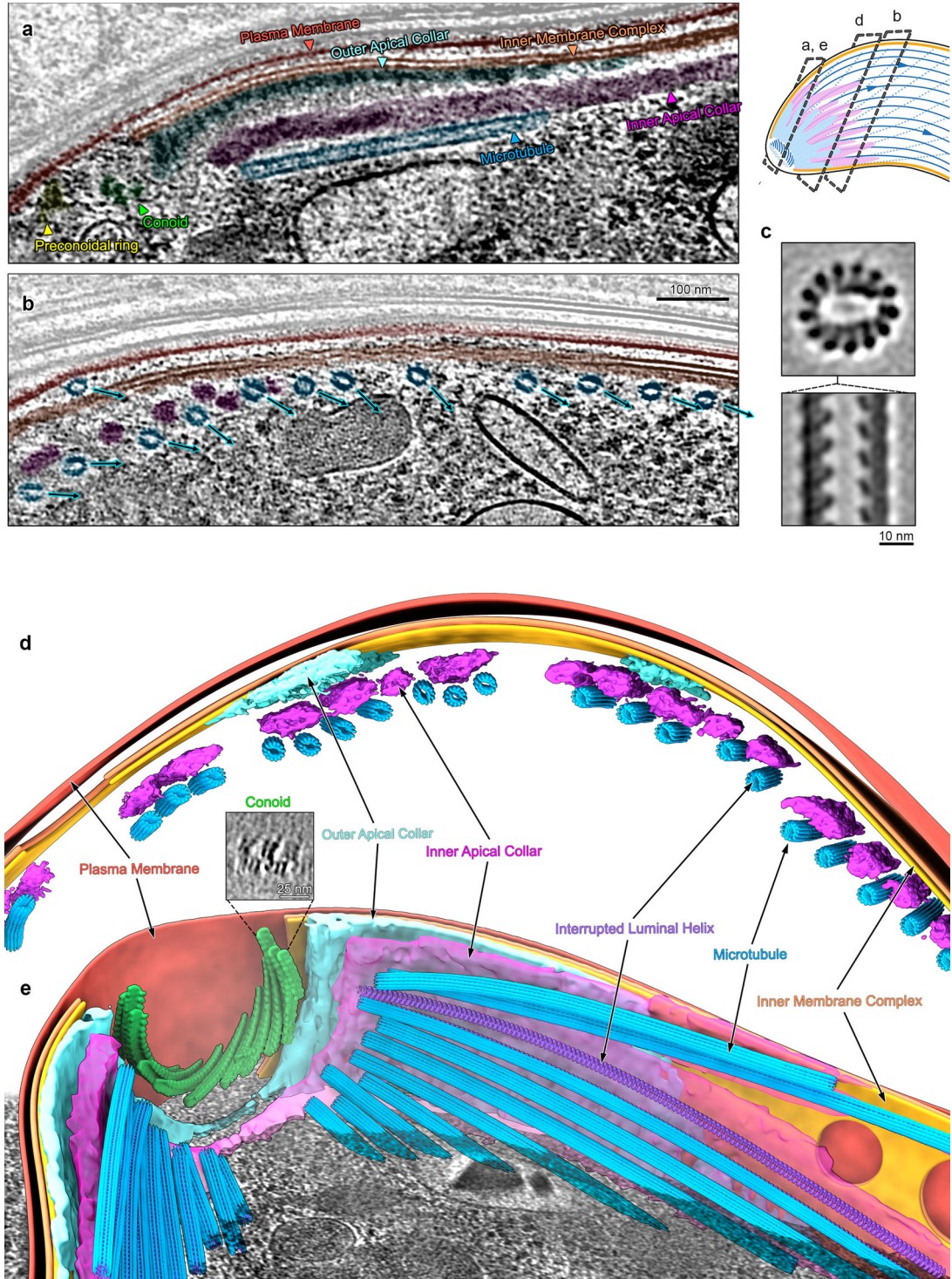

**Fig. 3 | Ookinete SPMTs contain interrupted luminal helices and are organised in a complex apical pole. a** Tomogram slice through the apical end of an ookinete with SPMTs oriented along the apico-basal axis. A cartoon of an ookinete on the right shows approximate centre positions of lamella in a, b and d. **b** Slice through an ookinete apex proximal region with SPMTs cut transversally, the annotation colours are the same as in a. SPMTs get closer to the IMC as inner apical collar tapers off (from left to right). The rotational orientation of SPMTs (centre to seam) is indicated with arrows. **c** Orthogonal sections through EM map determined by

SVA of ookinete SPMTs. **d** Segmentation of an apex proximal region of an ookinete with transversally cut SPMTs, highlighting the two apical collar layers between the SPMTs and IMC. The conoid is shown in green. **e** Segmentation of the apical pole of an ookinete. The tubulin density of one SPMT was hidden to reveal the ILH. Inset shows a slice through an average volume of the conoid periodic structure. Unannotated slice through tomogram shown in Fig. S3b and full volume in supplemental movie S2.

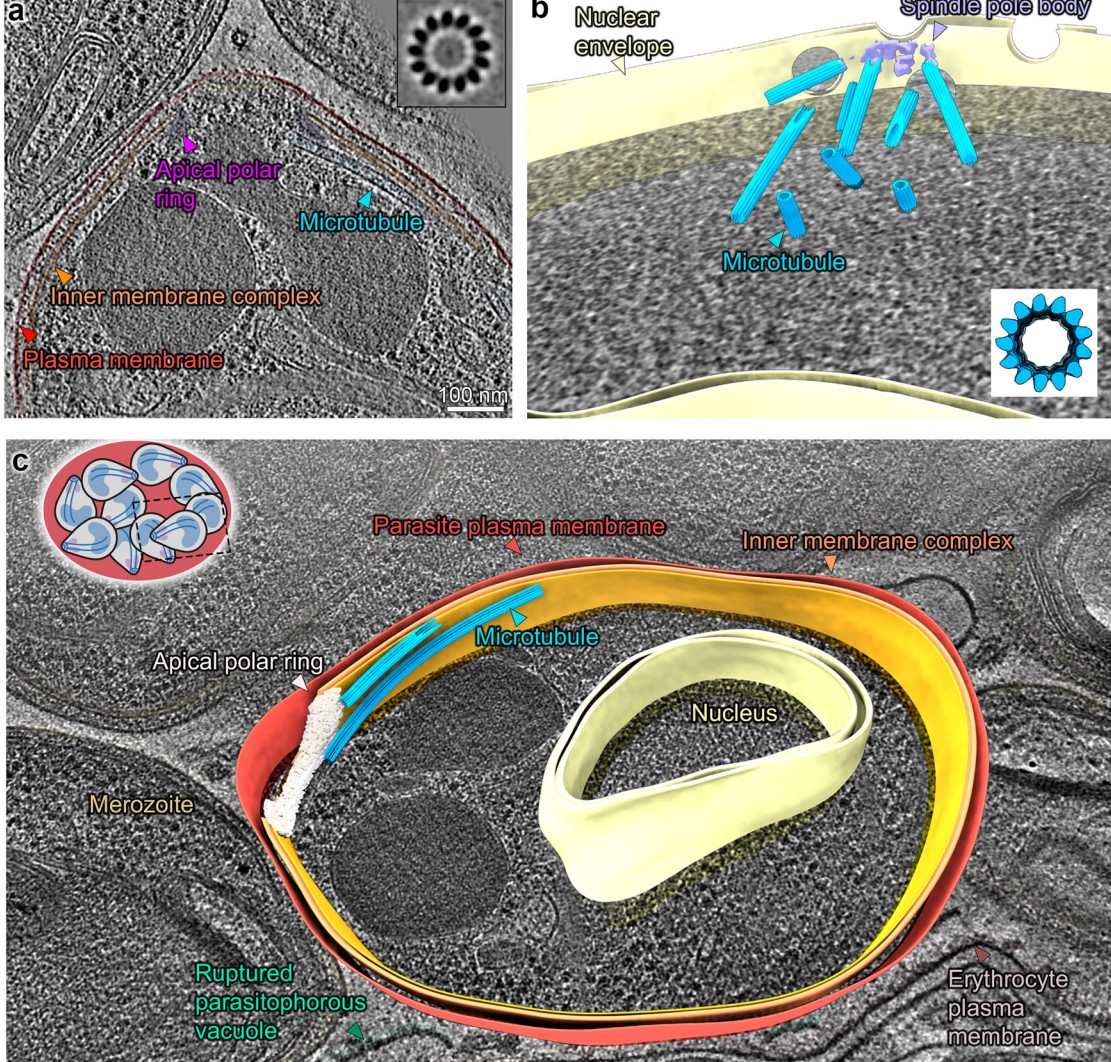

**Fig. 4 | Merozoites have canonical subpellicular (SPMTs) and nuclear spindle microtubules. a** Slice through a tomogram of an apical pole. Inset: slice through the microtubule subvolume average showing 13 protofilaments. **b** Segmentation of a nucleus in a dividing schizont with a partial spindle body. All four nuclear pore complexes observed in this section of the nucleus are in proximity of the spindle. Inset: slice through an isosurface of the SPMT EM map. **c** Segmentation of a single merozoite within a fully segmented schizont. Note: for simplification the IMC is segmented as a continuous double membrane although multiple discontinuities were observed. Unannotated slice through tomogram shown in Fig. S3c and full volume in supplemental movie S3.

characterised protease inhibitor (E64)[29], resulting in merozoites in erythrocyte membrane sacks.

Unexpectedly and in contrast to mosquito forms, there was no ILH in merozoite SPMTs (Fig. 4a, c). Analysis by SVA showed that merozoite SPMTs are canonical, i.e. made of 13 protofilaments. Similarly, spindle microtubules from dividing schizonts were canonical and lacked an ILH (Fig. 4b). However, in contrast to SPMTs which were uncapped, spindle microtubules contained a capping density at their minus ends which resembled γTuRC (Fig. S4c)[30]. This indicated that *Plasmodium's* microtubule cytoskeleton structures are altered dependent on the life cycle stage and that different populations of microtubules may be nucleated by distinct mechanisms.

### Gametocyte microtubules are non-canonical 13- to 18-protofilament singlets, doublets and triplets

After revealing that microtubule structures are different between the mosquito forms and asexual merozoite blood form, we next set out to study the SPMTs in the sexual gametocytes. The *P. falciparum* gametocyte develops from a round, to an elongated, and eventually falciform shape through five morphological stages. SPMTs start

polymerising in stage II and by stage IV they completely surround the cytoplasm. We enriched gametocytes at different stages in the maturation process and investigated their microtubules.

The pellicle coverage increased with subsequent developmental stages and with it the number of SPMTs in each cell. In all gametocyte stages, SPMT cross sections had noticeably large and variable diameters, ranging from 27 to 37 nm, and consisted of a mixture of singlets, doublets, triplets (often with unusual geometries), and even a quadruplet (Figs. 5, S5). This was unexpected as in all organisms studied to date, cytoplasmic microtubules are single tubes (singlet microtubules) and previous studies reported smaller diameter microtubules[31]. Doublets (canonically consisting of 13 + 10 protofilaments) and triplets are found in cilia and flagella[26,32], with triplets being hallmarks of centrioles. Consistent with merozoite SPMTs, we found no evidence of an ILH in these giant microtubules. SVA was performed independently on each of the ~170 singlet and ~30 A-tubules of doublet SPMTs and amazingly showed that they consisted of 13, 14, 15, 16, 17 or 18 protofilaments (Fig. 5c, Table S1). Although some canonical microtubules were present, these only accounted for 9% of the population, while 17-protofilament microtubules were the most

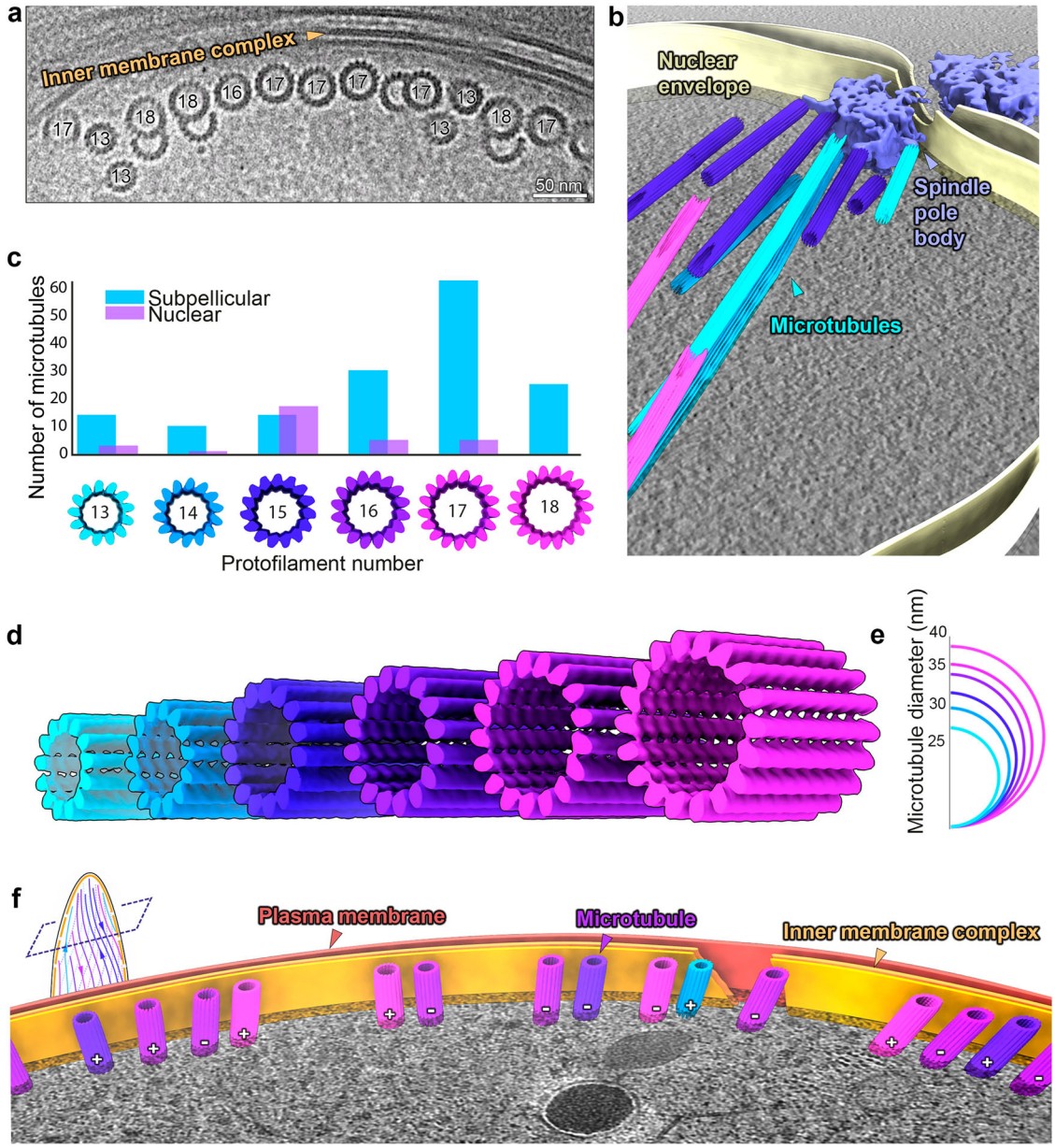

**Fig. 5 | Gametocyte subpellicular microtubules (SPMTs) have a wide range of protofilament numbers with random polarity. a** Micrograph showing a row of singlet and doublet SPMT cross sections and highlighting the range of sizes. Protofilament numbers are indicated (for doublets of the A tubule). Micrographs at two different tilt angles were stitched to show SPMT transversal views. **b** Segmentation of a stage III gametocyte nucleus with a spindle pole body at a nuclear pore complex. Microtubule colours correspond to protofilament numbers as shown in c, d. **c.** Bar chart of the distribution of different protofilament numbers in SPMTs (blue) $N = 155$ and nuclear spindles (lilac) $N = 31$. Distributions are significantly different ($p = 5 \times 10^{-8}$, chi-squared). **d** Isosurfaces of microtubules from subvolume averaging with protofilament numbers from 13 to 18. **e** Schematic representation of the differences in microtubule diameter. **f** Segmentation of a stage III gametocyte with transversely sectioned SPMTs. Microtubule colours correspond to protofilament numbers as shown in **c**, **d**. '+/−' indicates the polarity of each microtubule. Unannotated slice through tomogram shown in Fig. S3d and full volume in supplemental movie S4.

frequent (40%) (Fig. 5c). Doublets were also giant and their A-tubules had a similar distribution, with 17 protofilaments being the most common (Fig. S5b, c). Surprisingly, considering the ubiquity of a 13-protofilament A-tubule in other organisms, there were no A tubules with 13 protofilaments. The diversity in protofilament numbers sets the gametocyte SPMT population clearly apart from all other forms. Whether this is due to a gametocyte specific SPMT nucleation or a different mechanism could be investigated by comparison to microtubules in other cellular compartments.

Other than SPMTs, there are two more microtubule populations in gametocytes: nuclear spindle (or hemi-spindle[33]) and cytoplasmic. Cytoplasmic microtubules are short-lived and only appear in early

stage gametocytes on the opposite side of the cell from the developing pellicle[34]. Consequently, cytoplasmic microtubules are relatively rare. We observed a mixture of protofilament numbers (13 to 16, Fig. S5e) and while there was an indication of clustering by protofilament size, this could not be confirmed due to low numbers. Nuclear microtubules were seen frequently in gametocyte stage III/IV parasites and a full spindle pole body was present in one tomogram (Fig. 5b). Spindle microtubules ranged from 13 to 17 protofilaments, with 15 being most common (Fig. 5b, c). This was unexpected as, irrespective of protofilament number, all minus ends were clearly capped with a structure which could be γTuRC (although it is flattened relative to the canonical form). Notably, γTuRC has, so far, been exclusively described in

canonical microtubules (Fig. S4d). Together this data suggests that the wide distribution of protofilament numbers may be a result of differential isoform expression or post-translational modification, rather than a nucleation mechanism.

### Structurally diverse apical polar rings coordinate microtubules in *Plasmodium* invasive forms

Having observed substantial structural differences among microtubule populations between the four individual forms, we focused our analysis on the apical assemblies where nucleation of SPMTs occurs. In the invasive parasite forms (merozoites, sporozoites and ookinetes) proteinaceous rings (APR) at the apical pole act as unique MTOCs, coordinating SPMTs in addition to terminating the IMC. Consistent with this, all SPMTs in both mosquito forms and merozoites had the same polarity, with their minus ends at the apical pole and their plus ends reaching towards the basal cell pole. In all invasive forms, their minus ends were blunt and lacked a γTuRC cap, although there may be additional microtubule associated proteins (MAPs) in the lumen of ookinete minus ends (Fig S4a). However, despite all three forms containing 13-protofilament SPMTs with the same polarity, there were large-scale differences in the architecture of their apical poles (Figs. 2–4).

The merozoite's apex was the simplest, with only two to three microtubules radiating from the APR (Fig. 4c). Rather than being distributed equally around the APR, all SPMTs originated on the same side of the ring. Sporozoites contained a larger number of SPMTs. In two tomograms we could see full sporozoite APRs with a complete set of 13 closely spaced and a single opposing SPMT (Figs. 2e, f, S6). SPMT minus ends were in an apparent direct contact (Fig. S6a) and flush with the apical edge of the APR. However, each SPMT had a different angle-of-attack relative to the ring rotational symmetry axis which is tilted by roughly 45°[35] from the parasite apico-basal axis (Fig. S6c, d). The SPMT-APR contact, therefore, needs to allow a high degree of flexibility. In contrast, the SPMT radial orientation was constrained, with protofilaments 6 to 9 preferentially in contact with the APR (Fig. S6c).

The most elaborate apex was observed in ookinetes, with two concentric layers of amorphous protein, rather than a single ring (Fig. 3). Due to the large difference in morphology, we favour the term apical collar (AC)[15], but APR is equally appropriate due to functional similarity. Both rings separated into tentacles on their basal side (Fig. 3b, d, e), with the inner AC being in direct contact with and following individual SPMTs for up to ~1 μm. To accommodate the large number of SPMTs present in this form (~50–60) in the narrow apical end, the microtubule minus ends originated at staggered positions from the AC apical rim (Fig. 3e).

### Gametocyte SPMTS are nucleated independently from an APR

Gametocyte cells do not have clear cellular polarity, unlike the distinctly polar cells of the three invasive forms described above (Figs. 2–4). We observed no evidence of an APR-like structure or other structure that could act as a MTOC at gametocyte poles, and SPMTs originated at apparently uncoordinated positions along the cell length. Unlike in the invasive forms where all SPMTs had the same polarity with minus ends at the apical end, gametocyte SPMTs had an apparent random polarity (Fig. 5f, Table S1). Furthermore, there was no discernible pattern in the distribution of SPMTs along the IMC with respect to the number of protofilaments or secondary tubules (Fig. 5f, Table S1). Considering that SPMTs in the three earlier forms lacked a γTuRC cap, we had initially hypothesised that SPMT nucleation factors may be physically associated with the APR or AC. However, focused analysis of the gametocyte SPMT minus ends showed that they too were uncapped (Fig. S4). Together, this suggests a γTuRC-independent nucleation mechanism that is not uniquely associated with the APR.

### SPMT to IMC distance is conserved in all parasite forms, indicating a universal linker protein

Although *Plasmodium* parasites alter their microtubule structure, MTOCs, and higher order microtubule organisation at each step in their life cycle, one unifying feature is the interaction of SPMTs with the IMC. To elucidate whether the SPMT-IMC interaction is mediated in the same manner in all parasite forms, we measured the shortest distance ($d_{IMC}$) from the surface of each SPMT to the inner IMC membrane (Fig. 6). A similar $d_{IMC}$ in all parasite forms would suggest that the same proteins are involved in tethering SPMTs to the IMC. Excluding the apical assemblies, there was no significant difference in the $d_{IMC}$ between the four parasite forms (with a common median of 18 nm and standard deviation of 10 nm, Fig. 6d). This distance increased at the ookinete apex to ~50 and ~100 nm in order to accommodate the AC, but the distance from the SPMT surface to the closest surface, the inner AC, was comparable to $d_{IMC}$ (~13 nm, Fig. 6a). Thus, we show that, although each stage has a unique SPMT structure, the distance between the SPMTs and IMC remains consistent between stages. We therefore hypothesise that the same proteins link the SPMTs to the IMC in all parasite forms analysed.

### The interrupted luminal helix likely controls the rotational orientation of SPMTs with respect to the IMC

With the hypothesis that a common linker is tethering the SPMTs to the IMC in all forms, and with the knowledge that mosquito form SPMTs have a preferred radial orientation (Figs. 2e, 3b, S6b, c, S7), we set out to look for the linker binding site. A clear candidate, due to it being a distinct asymmetric structural feature, was the seam. We measured the angle, $\varphi_{IMC}$, between the seam, the microtubule centre and the closest point on the IMC surface (see supplemental information for details of seam orientation).

Analysing the mean $\varphi_{IMC}$ for all possible SPMTs revealed that only mosquito stage SPMTs have a set radial polarity. Surprisingly, the seam points away from the IMC with protofilament 8 oriented towards it instead (Figs. 6a, b, S6, S7). In contrast to the mosquito forms, no clustering of $\varphi_{IMC}$ was observed in gametocytes, suggesting that in this life cycle form the linker protein does not have a specific binding position (Fig. 6c).

Although these results did not find a universal binding site for the linker on all SPMTs, they suggested that the ILH was involved in setting the radial orientation with respect to the IMC. We inspected the mosquito stage SPMT EM density map for possible clues. Even at low contour levels there was no linking density between the SPMT and IMC, indicating that any linker protein is flexible or present at a low occupancy. Nevertheless, there was some evidence of radially-asymmetric decoration with an unknown protein density between protofilaments 10, 11 and 12, with additional density emanating radially outwards from protofilaments 11 and 12 (Fig. S8). If these densities correspond to a part of the SPMT-IMC linker, there would likely need to be additional SPMT-IMC connections (possibly at protofilament 6 or 7) to establish the observed radial orientation of protofilament 8 pointing towards the IMC. Thus, although the precise details of a linker are yet to be elucidated, we present evidence for asymmetric decoration and radial polarity, which are likely ILH-dependent, in the mosquito form microtubules.

## Discussion

The *P. falciparum* parasite has evolved a complex life cycle, moving through multiple different tissues within its human host and mosquito vector. For each new environmental niche, the parasite undergoes extreme morphological changes. As the parasite morphs into new specialised shapes for each niche, its microtubules are disassembled and subsequently reassembled into multiple unique, fit-for-purpose structures, coordinated by unusual MTOCs (Fig. 7). While the ookinete apical end is the most elaborate and includes a tubulin-based conoid,

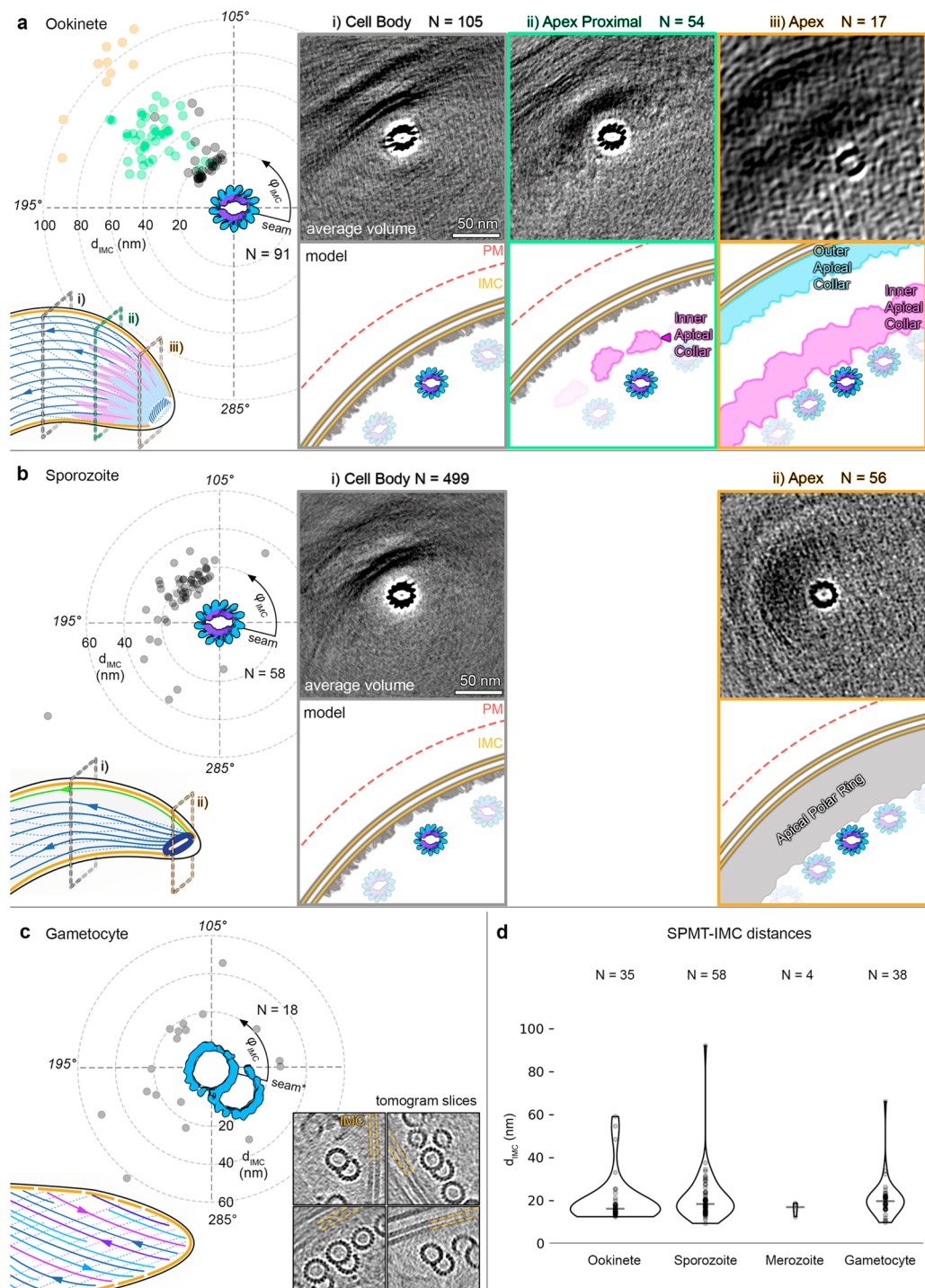

**Fig. 6 | Microtubule distance from the IMC is consistent in all forms, but the interrupted luminal helix may be needed to uniquely determine their radial orientation.** Panels **a**–**c** left: scatter plots of individual SPMT distance (dIMC) and radial orientation (φIMC) with respect to the IMC. Each point represents the median along a single SPMT. Ookinete and sporozoite SPMTs have a defined polarity whereas gametocyte orientations are random (see Fig. S7 for 2D histogram representation of these data). Parasite form cartoons indicate subcellular locations where data points were sampled: at the cell body or the apex. Right: average volumes of SPMTs sampled at the indicated subcellular locations. Although the subvolume averaging was focused on SPMTs, IMC and APR components can be

seen in the EM maps after extracting large subvolumes (200 nm edges). This is a consequence of the consistent SPMT-IMC distance and orientation. Cartoons below average volumes are models of the pellicle architecture at each location. Note that since the IMC wraps around the torus-shaped sporozoite APR, it was not possible to measure a single apical dIMC (apical φIMC angles are shown in Fig. S6b). Gametocyte panel (**c**) shows sections through individual SPMTs rather than an average, to indicate their inconsistent orientations and different number of protofilaments. *See supplementary materials for details and assumptions made in gametocyte orientation assignment. **d** Violin plot comparing dIMC between forms, widths are scaled according to the amount of data.

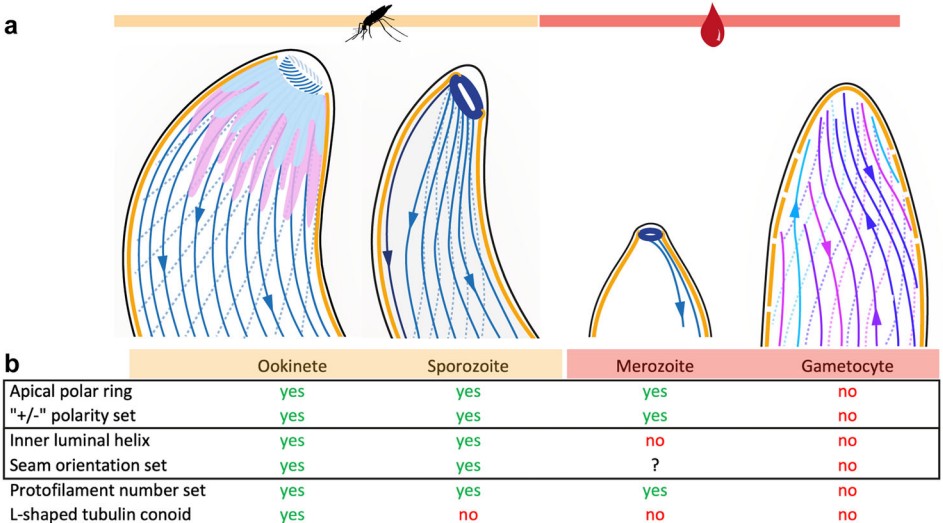

**Fig. 7 | Structural diversity of *Plasmodium* microtubules across its life cycle.**
**a** Cartoon representations of the four *Plasmodium* forms analysed. **b** Table summarising the main architectural differences in the four forms. Solid outlines indicate properties that are likely correlated: the presence of an apical polar ring sets the SPMT polarity and ILH is directly linked to setting the seam orientation with respect to the IMC.

| | Ookinete | Sporozoite | Merozoite | Gametocyte |
|---|---|---|---|---|
| Apical polar ring | yes | yes | yes | no |
| "+/-" polarity set | yes | yes | yes | no |
| Inner luminal helix | yes | yes | no | no |
| Seam orientation set | yes | yes | ? | no |
| Protofilament number set | yes | yes | yes | no |
| L-shaped tubulin conoid | yes | no | no | no |

gametocytes stand out among the other forms with their lack of an APR and diverse population of microtubule structures (Figs. 5, 7).

Gametocytes have a large range and variety of microtubule structures including singlets with 13 to 18 protofilaments and giant doublets, triplets and quadruplets. Such large structural diversity is so far unprecedented, which raises questions about the underlying mechanism and the physiological role. According to our current understanding, protofilament number is tightly controlled, but this apparently isn't the case in gametocytes. Microtubules have the propensity to self-assemble in vitro, albeit at concentrations considerably higher than physiological. This spontaneous self-assembly results in microtubule populations with a wide distribution of protofilament numbers (9–16 for bovine brain tubulin)[1,16]. However, a nucleation-free self-assembly is unlikely to be responsible for the giant non-canonical SPMTs in gametocytes as their polymerisation is restricted to the surface of growing IMC plates, rather than spontaneously in the cytoplasm. Furthermore, we occasionally observed groups of cytoplasmic microtubule populations, possibly clustered according to diameter (Fig. S5e) which suggests that a level of control is available.

An increased number of protofilaments could be due to altered post-translational modifications or the expression of a novel MAP or tubulin isoform. *Plasmodium* spp. express one β- and two α-tubulin isoforms. Although α1- and α2- tubulin exhibit ~95% sequence identity, there are several amino acid substitutions at key regions. α2- tubulin, for example, is missing 3 amino acids at its C-terminus (often a target for posttranslational modifications) and the least conserved region in the sequence corresponds to a loop which may mediate intra-protofilament interactions. These small changes could ultimately result in alterations in the microtubule lattice and promote B-tubule formation leading to doublets and triplets[36]. Furthermore, previous studies into tubulin isoforms in humans have shown that the relative fraction of an isoform within a microtubule can modulate the microtubule lattice[37], providing a possible explanation for the range in protofilament numbers observed. Although α2- tubulin is likely expressed in all life cycle forms, its relative expression is highest in sexual stages[38,39]. This isoform was not present in microtubules purified from asexual stages[40], is not essential in asexual forms[41], and can only partially replace α1-tubulin in *P. berghei* sporozoites[42].

*P. falciparum* is the only *Plasmodium* species that produces elongated gametocytes[43], and their SPMTs have been shown to be polyglutamated, which may play a role in their stability[28,44]. The maturing gametocyte has decreased deformability[45] which plays a role in inhibiting premature release of gametocytes from sequestration sites in the bone marrow, therefore avoiding splenic clearance[46]. Gametocytes are at their most rigid at stage IV, when all microtubules are assembled. However, the cells become deformable again in stage V, when the majority of the SPMTs have been disassembled, hinting at a link between cell rigidity and SPMTs.

We expect that giant microtubules will be significantly stiffer (less likely to bend) than the canonical form. The transition from 13 protofilaments to 17 or 18 protofilaments is expected to greatly increase the stiffness of an individual microtubule, as a change from 13 to 15 protofilaments was predicted to result in a 35% increase in stiffness[47]. This increased rigidity is evidenced in, for example, mechanosensory cells where their 15-protofilament microtubules are implicated in mechanotransduction. As the gametocyte periphery has a limited surface area, a limited number of microtubules can fit onto the inner surface of the IMC. Therefore, maximising the number of microtubules as well as assembling larger, more rigid microtubules may allow *P. falciparum* gametocytes to grow into their extended, rigid form and develop undisturbed in their human hosts until mature.

The two mosquito forms, on the other hand, have their own unique SPMTs made of 13 protofilaments and characteristically reinforced with an ILH. Our observations contribute to a growing list of organisms that make use of an ILH in their microtubule cytoskeleton, which now includes the SPMTs of apicomplexans[14,27] and the cilia and flagella of some mammals[25,26]. With our observations it seems likely that ILH consisting of TrxL1- and SPM1-like proteins are conserved across the *Apicomplexa*, but only in some life cycle forms. The TrxL1 expression profile in published data is consistent with our observation, with its transcription strongly upregulated in ookinetes and sporozoites and very little relative expression in blood stages[38,48]. In our TrxL1-GFP and SPM1-GFP lines, we saw high levels of expression in sporozoite (Fig. S2). Intriguingly, in mammals, there is a homologue of TrxL1 that localises to lung cilia and sperm flagella (thioredoxin-like Txl-2[49]) and a homologue of SPM1 in axonemes (stabilizer of axonemal microtubules SAXO2[50]), suggesting that the ILH may be a common eukaryotic trait.

The ILH has been hypothesised to limit turnover at the microtubule plus end and strengthen SPMTs in both the longitudinal (via SPM1) and transversal direction (TrxL1)[25,26]. An SPM1-null mutant in *T. gondii* had reduced fitness[12]. This was not seen in a second study,

although microtubules were more susceptible to chemical treatments when missing SPM1 or TrxL1[27]. It is conspicuous that ILH occurs in locomotive structures, such as cilia, flagellar ends and apicomplexan gliding forms. Thus, we postulate that the ILH provides stability without sacrificing flexibility, and if breakages occur, SPM1 keeps the ends together, allowing SPMT repair or self-healing. Where solely rigidity is required, *Plasmodium* gametocytes have evolved a dense sheet of SPMTs with a large number of protofilaments, doublets and triplets.

The function of SPMTs as cytoskeletal elements relies on a physical connection to the IMC. Studying the linkage between the SPMTs and the IMC, we found that the rotational orientation of ILH-containing SPMTs with respect to the IMC was set. Surprisingly, the seam was not oriented towards the IMC. This is consistent with and clarifies previous data on seam orientation from flattened detergent solubilised *T. gondii*[14], but remains surprising as the seam is the only truly asymmetric microtubule feature that can be accessed by externally binding MAPs. Instead, the ILH could be setting the rotational orientation directly. Gametocytes, which lack an ILH, have an apparently random radial polarity. Our evidence for this was the large variance of doublet $\varphi_{IMC}$, but beyond this it is also difficult to reconcile the supertwist of non-canonical microtubules with a linker having a preference for a specific protofilament (as different protofilaments are directed towards the IMC as the microtubule twists). We propose that the radial polarity of 13-protofilament SPMTs in merozoites is also random, due to the supposed lack of an asymmetric feature like the ILH.

There are two conceivable mechanisms for how the ILH could be orienting SPMTs. Either a direct contact between an ILH component and an external MAP, or by forcing SPMTs into elliptical cross sections (Fig. S1), where the inconsistent inter-protofilament angles could be recognised as binding sites. We observed weak densities emanating from protofilaments 11 and 12 and in the ridges between protofilaments 9 to 12 (Fig. S8). These densities are not consistent with known MAPs such as kinesin or doublecortin, but the resolution of our C1 symmetry EM density maps is not high enough to be conclusive. It is not clear whether they correspond to a putative SPMT-IMC linker, considering that they are not oriented towards the closest point on the IMC. A framework for understanding the evolution of asymmetric rotational orientation of SPMTs may be a recently proposed hypothesis that SPMTs originate from an ancient flagellum[51,52]. The orientation of ILH containing SPMTs is roughly consistent with the orientation of the A-tubule in axonemes with respect to the membrane. Whether this radial polarity is important for orchestration of directional gliding will need to be tested.

The variations in microtubule structures among *Plasmodium* forms pose the obvious question; how are these different microtubules being nucleated? It has been suggested that SPMTs with ILH could be nucleated by SPM1[27]. TrxL1 could then set the protofilament number to 13 through the curvature of its oligomeric state. This seems plausible in isolation, but not in light of our new data. It does not explain how the giant microtubules of gametocytes or merozoite 13-protofilament SPMTs, which lack the ILH, are nucleated. Furthermore, the curvature of TrxL1-bound protofilaments departs from that of a canonical 13-protofilament microtubule (Fig. S1) and there is a gap in the ILH between protofilaments 13 and 1 where additional protofilaments could conceivably be inserted. γTuRC is an unlikely nucleation mechanism as SPMT minus ends were uncapped in all microtubules observed from all forms (Fig. S4). Although it is possible that the γTuRC caps could have been disassembled (γ-tubulin is expressed in schizonts, for example[53]), this would be surprising due to the stability of *Plasmodium* microtubules. It is difficult to speculate on the nature of giant microtubule nucleation in gametocytes. On one hand one could speculate about a γTuRC-independent nucleation of SPMTs that allows protofilament numbers from 13 to 18. But on the other, is the presence of a γTuRC-like structure on spindle microtubules, which nevertheless is unable to generate a homogeneous microtubule population. Recent

data suggests that, at least initial SPMTs in gametocytes, are nucleated at a nuclear membrane-located centriolar-plaque[44]. How different diameter microtubules and our observed random polarity of SPMTs can be achieved with this model will still need to be explored. Together, these data may hint at a novel SPMT nucleation mechanism and with it, a potential target for antimalarials.

In this work we reveal the metamorphosis of microtubule structures throughout the life cycle of *P. falciparum*. On the highest level, the remarkable structural diversity forces us to re-evaluate our preconceptions about the canonical microtubule resulting from the overreliance on a small number of model organisms. The high resolution and optimal structural preservation of the in situ data presented here provide a framework for integrating findings from other, more reductionist methods, in order to gain a complete picture of an organism's cytoskeleton. We expect that as more cell types are studied, using these techniques available now, the true diversity of eukaryotic microtubule structures and their niche functions will be larger than currently anticipated.

## Methods

### Cell culture
*P. falciparum* blood forms were cultured in human red blood cells (O + or B + , Universitätsklinikum Eppendorf, Hamburg, Germany) at 5% haematocrit, 1% $O_2$, 5% $CO_2$, and 94% $N_2$ at 37 °C, in RPMI complete medium containing 0.5% Albumax II[54].

### Parasite synchronisation
Schizonts were harvested using 60% prewarmed Percoll[55], washed once with prewarmed RPMI, resuspended in 5% uninfected red blood cells in prewarmed RPMI and cultured for 3 h, until parasites egressed and reinvaded. Unruptured schizonts were lysed with 5% sorbitol. The remaining culture contained young rings with a 3-h synchronicity window.

### *P. falciparum* schizonts
Tightly synchronised 3D7 schizonts expressing endogenously GFP-tagged GAPM2 (glideosome-associated protein with multiple membrane spans 2[56,57]) were isolated from a 10–20 mL culture (5–10% parasitaemia) using 60% Percoll[55], washed twice in pre-warmed RPMI, and treated with 1 μM of the PKG-inhibitor compound 2 (provided by Dr. Mike Blackman, The Francis Crick Institute, UK)[58,59], and incubated for 6 h. Segmented schizonts were washed once and then resuspended in pre-warmed RPMI without Albumax or phenol red but with the addition of 1 μM E64 (Sigma) to allow the parasitophorous vacuole to rupture but prevent rupture of the red blood cell membrane. This helped us identify very late stage schizonts in the SEM. Cells were kept warm and vitrified within an hour.

### *P. falciparum* gametocytes
Gametocyte stages were generated by targeted overexpression of the sexual commitment factor GDV1 (Gametocyte development 1) using a 3D7 inducible gametocyte producer line (iGP) as previously described[60] overexpressing a GFP-tagged version of the suture inner membrane complex protein PF3D7_1345600 (plasmid from[57]). GDV1-GFP-DD expression was achieved by the addition of 2 or 4 μM Shield-1 to a parasite culture containing 2–3% rings. Shield-1 was maintained for a further 48 h until reinvasion. The sample was adjusted to 10% parasitaemia and cultured in RPMI supplemented with 10% human serum (blood group AB+) for 10 days to allow gametocyte maturation. To deplete non-committed asexual forms, gametocytes were treated with 50 mM *N*-acetyl-d-glucosamine (GlcNac) for at least 4 days. Culture medium was changed at least once per day on a 37 °C heating plate. Gametocyte stages II, III, IV and V were isolated at different time points during gametocyte maturation from a 20 mL culture using 60% prewarmed Percoll. Isolated gametocytes were washed twice and then

resuspended in pre-warmed RPMI without albumax, serum or phenol red and kept warm (for between 15 min and 45 min) until immediately prior to freezing.

### *P. falciparum* sporozoites

*P. falciparum* sporozoites (strain: NF54-ΔPf47-5′*csp*-GFP-Luc: expressing a GFP-Luciferase fusion protein) under the control of the csp promoter, genomic integration, no selection marker[61] were prepared at TropIQ (Nijmegen, Netherlands).Gametocytes were fed to 2 days old female *Anopheles stephensi* mosquitoes. Mosquito infection was confirmed 7 days post infection by midgut dissection. At 7 days post infection, the infected mosquitoes received an extra non-infectious blood meal to boost sporozoite production. Two weeks post infection, sporozoites were isolated using salivary gland dissection and shipped to Hamburg at room temperature.

### *P. berghei* ookinetes

Ookinetes were produced using the *P. berghei* line PbRFP, a *P. berghei* ANKA line that constitutively expresses RFP. Two female Swiss mice were injected intraperitoneally with 200 μl phenylhydrazine (6 mg/ml in PBS) to stimulate reticulocytosis. Two days later, the mice were infected intraperitoneally with 20*10⁶ iRBC PbRFP. Mice were bled three days post infection and 500 μl blood was transferred to 10 ml ookinete medium (RPMI supplemented with 20% (v/v) FCS, 50 μg/ml hypoxanthine, and 100 μM xanthurenic acid, adjusted to pH 7.8 – 8.0) at 19 °C. After 22 h of culture, ookinete cultures were underlaid with 5 ml 55% Nycodenz/PBS and centrifuged for 25 min at 210 xg without brake. The interphase containing purified ookinetes was collected, washed in PBS and immediately plunge frozen for EM.

### Plunge freezing

Cells were kept warm in a 37 °C heat block next to the plunge freezer. 3 μl of enriched parasites were applied onto a freshly plasma-cleaned UltrAuFoil R1.2/1.3 300 mesh EM grid (Quantifoil) in a humidity controlled facility. Excess liquid was manually back-blotted and grids were plunged into a reservoir of ethane/propane using a manual plunger. Grids were stored under liquid nitrogen until imaging.

### Cryo FIB milling

Grids were clipped into autogrids modified for FIB preparation[62] and loaded into either an Aquilos or an upgraded Aquilos2 cryo-FIB/SEM dual-beam microscope (Thermofisher Scientific). Overview tile sets were recorded using MAPS software (Thermofisher Scientific) before being sputter coated with a thin layer of platinum. Good sites with parasites were identified for lamella preparation before the coincident point between the electron beam and the ion beam was determined for each point by stage tilt. Prior to milling, an organometallic platinum layer was deposited onto the grids using a GIS (gas-injection-system). Lamellae were milled manually until under 300 nm in a stepwise series of decreasing currents. Milling was performed at the lowest possible angles to increase lamella length in thin cells. Finally, polishing of all lamella was done at the end of the session as quickly as possible but always within 1.5 h to limit ice contamination from water deposition on the surface of the lamellae. Before removing the samples, the grids were sputter coated with a final thin layer of platinum. Grids were stored in liquid nitrogen for a maximum of 2 weeks before imaging in the TEM.

### Cryo-EM

FIB-milled grids were rotated by 90° and loaded into a Titan Krios microscope (Thermofisher) equipped with a K2 or K3 direct electron detector and (Bio-) Quantum energy filter (Gatan). Tomographic data was collected with SerialEM[63] with the energy-selecting slit set to 20 eV. Datasets were collected using the dose-symmetric acquisition scheme at a ± 65° tilt range with 3° tilt increments. For all datasets, 5–10 frames were collected and aligned on the fly using SerialEM and the total fluence was kept to less than 120e⁻/Å². Defoci between 3 and 8 μm underfocus were used to record the tilt series'.

### Tomogram reconstruction

Frames were aligned on the fly in SerialEM; CTF estimation, phase flipping and dose-weighting was performed in IMOD[64]. Tilt-series' were aligned in IMOD either using patch-tracking or by using nanoparticles (likely gold or platinum) on lamella surfaces as fiducial markers. Tomograms were binned 4x and filtered in IMOD or by using Bsoft[65].

### Sub-volume averaging

**For all averages.** SVA was performed in PEET[66] with tomograms at progressively lower binning. Initial 3D coordinates (model points or particle coordinates) for SVA were generated by interpolating between manually traced microtubule centres in IMOD, using scripts based on TEMPy[67] and using Python libraries Numpy, Scipy and Matplotlib[68–70]. The vectors between pairs of points were used to set initial particle Y axis orientation (microtubule pseudosymmetry axis). Each microtubule was processed separately, aligned to a unique reference (raw average) generated by averaging the respective particles with initial orientations (Fig. S9). In order to determine its polarity and the number of protofilaments, the initial rotations around the Y axis (here referred to as φ rotation) were randomised (Fig S9, step 2). Particles from the same forms with the same number of protofilaments were then combined for further processing. The progress of SVA was monitored by inspecting pin models in UCSF Chimera (Fig. 2e) where each particle's position and orientation were represented by a pair of markers. This allowed us to e.g.: identify and remove outliers from the linear geometry. Particles were pruned if they overlapped with others, had low cross-correlation coefficient or drifted too far from their initial positions. SPMT maps were sharpened using arbitrary B-factors using Bsoft[65].

### Sporozoite and ookinete SPMTs

Having determined the relative polarity, the necessary step for combining is finding the relative φ rotation (rotation around the symmetry axis) between individual microtubules. This could be done by aligning each individual particle to a common reference, but would result in a large number of errors due to the low signal to noise ratio. We developed a method analogous to that of Zabeo et al.,[25] where the microtubule average volumes are aligned together in order to find their relative φ rotation (Fig. S9 step 4). These φ rotations were then applied to their respective particles and aligned to a common reference. SVA was subsequently performed with volumes binned 4 and 2 times. Only sporozoite data were aligned with unbinned volumes; the final step was to replace volumes reconstructed with the whole sets of tilts by volumes reconstructed with tilts between ±24°. No rotation search was performed with the restricted tilt range. The resulting C1 EM map was anisotropic around the pseudosymmetry axis due to an uneven distribution of microtubule rotational orientations. To address this, particles were separated into classes by orientation and particles with the lowest cross correlation coefficients in the most abundant classes were removed. This reduced the number of particles from 24028 to 13263 in sporozoite and 8377 to 1851 in ookinete datasets.

### Other microtubules

**Gametocyte and merozoite singlet microtubules.** Approximately 200 microtubules from 25 tomograms were processed individually, with random initial φ rotations. Data were then manually classified by protofilament number, rotated to the same polarity and aligned with volumes binned 4 and 2 times. Helical parameters were measured directly using class averages with the exception of 13- and 15-protofilament classes where tubulin subunits were not resolved and

**Table 1 | Oligonucleotides used for cloning and genotyping PCR**

| Primer-ID | Primer name | 5' → 3' sequence |
|---|---|---|
| Cloning primers | | |
| P2229 | SPM1-fw-EcoRI | CCGGAATTCATGGAAATAATAGGCGCAAAAC |
| P2230 | SPM1-rev-6Ala-BamHI | CGCGGATCC**CGCCGCCGCCGCCGCCGC**ATACCACGCTTTTTTTACATCATC |
| P2233 | TrxL1-fw-EcoRI | CCGGAATTCATGTCTTGTGCTAATTTTAATTCCC |
| P2234 | TrxL1-rev-6Ala-BamHI | CGCGGATCC**CGCCGCCGCCGCCGCCGC**TAAACCTTTTTTATTTAAAGAATAAATATTATTTCTG |
| Genotyping primers | | |
| P210 | TestprimerGFPreverse | TTAACATCACCATCTAATTCAACAAG |
| P650 | QCR2_81070 | ACGTGCATTTCTTAGCGTTCCT |
| P862 | GFPtestR | TCCAGTGAAAAGTTCTTCTCCT |
| P2219 | PBANKA_082020_GT | AGCGCGCATTAGCCAATTCT |
| P2221 | PBANKA_082020_QCR2 | ACGTTCTCCACATTGGCAAA |
| P2224 | PBANKA_081O7OO_GT2 | CACAACACATAAAAAATGCGCACC |
| P2239 | pL18-tgdhfr-seq-AB | ACTTTAGAGGCCATGAAGAG |

Bases encoding for the 6-alanine linker are in blod.

published parameters were used (Table S2)[71]. Helical symmetry was applied, followed by SVA alignment.

**Gametocyte doublet microtubules.** SPMT doublets were processed in a manner analogous to ookinete and sporozoite data, where average volumes were used to determine relative φ rotations. SPMTs with different numbers of protofilaments in A and B tubules were combined together, resulting in an ensemble average volume.

**Half map generation.** Particles were split into two halves (even and odd numbered particles). Random coordinate offsets of up to ±2 nm and random angular offsets of up to ±3° were added to each particle and the resulting parameters were used to re-generate a raw average for each half data set. Alignment of the two halves was then done independently with twice binned and then unbinned volumes, following the same procedure as the whole dataset. Fourier Shell Correlation was measured using Bsoft.

**Structure visualisation.** EM maps and atomic models were visualised with UCSF (University of California San Francisco) Chimera[72] or UCSF ChimeraX[73]. Computational sections were generated in IMOD.

**Multiple sequence alignment.** ClustalOmega[74,75] was used to align TrxL1 protein sequences and JalView was used for visualisation[76]. Colours are based on ClustalX colouring.

**Tomogram segmentation for visualisation.** Segmentation was performed manually in IMOD, using drawing tools followed by linear interpolation. The resulting models were used to extract segmented volumes. SPMTs, ookinete conoid and sporozoite APR were back-plotted: average volumes were placed into 3D volumes using coordinates determined by SVA. SPMT and APR particles were spline fitted to smooth alignment errors for visualisation. Segmented and back-plotted volumes were visualised using UCSF ChimeraX[73].

**Tomogram segmentation for IMC-MT distance and angular measurements.** Tomograms were bandpass filtered using Bsoft. Segmentations of the filtered tomograms were guided by a tensor voting algorithm using TomoSegMemTV[77,78]. The parameters were optimised for each dataset, respectively. Clusters containing the IMC segmentations were manually extracted by visual analysis. The clusters were then converted to a 3D point cloud and further processed using the Open3D library[78]. Statistical outlier analysis was used to remove excess noise from the segmentations. Subsequently, the DBSCAN algorithm was used to separate individual membrane sections and the outer

side of the IMC was selected manually for subsequent distance measurements. The angle φ_IMC was measured between two vectors for each SPMT particle: a vector of the SPMT particle X axis and a vector from the particle centre to the nearest segmented IMC coordinate (roughly equivalent to IMC normal vector intersecting the SPMT particle). Particles outside segmented membrane patches were excluded. Measurements from individual microtubules were reduced to a median value for plotting and statistical analyses.

The requirement for a clear membrane density for automated segmentation substantially reduced the number of microtubules available for analysis. Thus, due to the rarity of SPMTs in merozoites, doublets in gametocytes and minus termini in ookinetes, a small number of tomograms were segmented manually in IMOD.

**Atomic model alignment and fitting.** Initially, *T. gondii* SPMT model (PDB 7MIZ) was fitted to the *P. falciparum* sporozoite SPMT EM map as a rigid body in Chimera[72], the fit was then visualised in ChimeraX[73]. The resulting fit was unambiguous with TgTrxL2 (which is not expressed in *Plasmodium* but is part of the *Toxoplasma* ILH) outside and the remaining protein chains inside the map. Due to the difference in ellipticity between the *P. falciparum* and *T. gondii* maps, protofilaments and TrxL1/SPM1 half arcs were fitted as separate units. Structural prediction of PfTrxL1 (https://alphafold.ebi.ac.uk/entry/Q8I2W0)[79] was aligned with TgTrxL1 (PDB 7MIZ) using matchmaker[80] in ChimeraX.

**Protofilament angle measurements.** Inter-protofilament angles were determined by measuring the angle between Cα atoms of the same pair of residues in neighbouring protofilaments. Median values of roughly 200 measurements were used.

***Plasmodium berghei* parasite line generation.** Endogenous tagging was performed essentially as described before using single-crossover integration[81]. As both SPM1 and TrxL1 show an overall gene length of less than 1 kilobase, the entire open-reading frame was amplified from *Pb*ANKA wildtype genomic DNA. The resulting DNA fragment was cloned into the pL18 vector[82] using EcoRI and BamHI restriction sites. The reverse primer encoded for six alanines that were used as a linker. The pL18 vector harbours the *hDHFR* gene for positive selection using the drug pyrimethamine. Prior to transfection, the vector was linearized using SwaI (SPM1) and BsmI (TrxL1), respectively, followed by ethanol-precipitation. All oligonucleotides used to generate DNA fragments as well as those used for genotyping PCRs are listed in Table 1.

The linearized pL18-SPM1-GFP/pL18-TrxL1-GFP vectors were each transfected into an unmodified *P. berghei* ANKA strain using standard protocols[83]. Parasites that integrated the desired DNA construct were

selected by oral administration of pyrimethamine (0.07 mg/ml) via the mouse drinking water one day post transfection. Once mice showed between 1–3% infected red cells, blood was collected via cardiac puncture from anesthetized mice (100 mg/kg ketamine and 3 mg/kg xylazine, Sigma-Aldrich). For permanent parasite line storage, aliquots of 100 μl whole blood mixed with 200 μl freezing solution (10% Glycerol in Alsever's solution, Sigma-Aldrich) were stored in liquid nitrogen.

To verify correct integration, genomic DNA was isolated from whole blood and tested for correct construct integration by genotyping PCR. For this, erythrocytes were first lysed in 1.5 ml phosphate buffered saline (PBS) containing 0.03% saponin. After centrifugation and two washing steps, the genomic DNA was isolated using the Blood and Tissue kit (Qiagen Ltd) according to the manufacturer's protocol. Parasites from these transfections resulted in mixed (parental) populations were used for mosquito infections.

Plasmids and oligonucleotides were designed using SnapGene Software Version 3.2.1 (Insightful Science, available at snapgene.com). Images were analyzed with Fiji (Version: 2.0.0 rc 64/1.51 s)[84].

**Mosquito infection.** Frozen (non-clonal) parasite stocks were thawed and directly injected intraperitoneally (100–150 μl) into one mouse (6–8-week-old female Swiss mice) with the infection monitored by blood smears. Once the infected mouse reached 2–3% infected red cells, the mouse was anesthetized (100 mg/kg ketamine and 3 mg/kg xylazine, Sigma-Aldrich), bled and 20 million parasites transferred into two naïve recipient mice by intraperitoneal injection. Three days after fresh blood transfer mice were anesthetized (120 mg/kg ketamine and 16 mg/ml xylazine) and placed on a cage containing around 200 female *Anopheles stephensi* mosquitoes, which were starved for 3 to 5 h by removal of sugar and salt pads. Mosquitoes were allowed to feed on mice for 15 min to 30 min. Following infection, mosquitos were kept at 21 °C and 70% humidity.

**Sporozoite isolation and fluorescent imaging of live *P. berghei* sporozoites.** Salivary gland sporozoites were isolated on day 19 post mosquito blood meal. To this end, salivary glands were dissected on ice in RPMI and crushed with a pestle. Subsequently the tube was filled up to 1 ml with RPMI and the solution carefully underlaid with 3 ml of 17% Accudenz (Accurate chemical & scientific cooperation). Centrifugation at 1600 xg for 20 min at room temperature separated sporozoites from cell debris. The sporozoite containing interphase was collected (total volume of 1.4 ml) and sporozoites were spun down for 3 min at 100 xg (Thermo Fisher Scientific, Biofuge primo). Resuspension of pelleted sporozoites in RPMI containing 3% bovine serum albumin (ROTH) resulted in activated sporozoites. The resulting mixture was transferred into a well of an optical bottom 96-well plate (Thermo Fisher Scientific) and the plate was centrifuged for 3 min at 200 xg (Multifuge S1-R, Heraeus). Sporozoites were imaged using an epifluorescence microscope (CellObserver, Zeiss) and a 25× (NA 0.8, water) objective with an exposure of 80 ms. Images were taken until 1 h post sporozoite activation.

**Fluorescence microscopy of all other stages.** Fluorescence images were taken of live parasites on a Leica D6B fluorescence microscope equipped with a Leica DFC9000 GT camera and a Leica Plan Apochromat 60× or 100×/1.4 oil objective. Contrast and intensities were linear adjusted to present clear parasite shapes using Fiji and cropped images were assembled into a composition of cells for Fig. 1 using Adobe Illustrator CC 2021.

**Statistics and Reproducibility.** Five preparations of *P. falciparum* gametocytes were generated of which six grids were imaged, five preparations of *P. falciparum* schizonts were generated of which six grids were imaged, two preparations of *P. falciparum* sporozoites were generated of which four grids were imaged and two preparations of *P. berghei* ookinetes were generated of which three grids were imaged.

**Reporting Summary**
Further information on research design is available in the Nature Portfolio Reporting Summary linked to this article.

## Data availability
The subvolume averages generated in this study have been deposited in the EMDB under accession codes Sporozoite MT (EMD-15532), Gametocyte MTs: 13pf (EMD-15534), 14pf (EMD-15535), 15pf (EMD-15536), 16pf (EMD-15537), 17pf (EMD-15538), 18pf (EMD-15539). The distance measurement data generated in this study are provided in the Source Data file. All strains and plasmids are available upon request. Source data are provided with this paper.

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

## Acknowledgements

We would like to thank the CSSB Cryo-EM and ALFM facilities for their support. We thank Till Voss for the 3D7-iGP parasites, TropIQ for providing the *P. falciparum* sporozoites and Andrew Waters and Katie Hughes for sharing the PbRFP line. Thanks to Lindsay Baker for critical feedback on the results and John Heumann and Carolyn Moores for helpful discussions and advice. Thanks to the Stackoverflow community for being a seemingly bottomless and sanity restoring repository of knowledge. This work was funded by: HFSP long-term postdoctoral fellowship LT000024/2020-L (JLF), Infrastructures for the control of vector-borne diseases (Infravec2) funded by the EU's Horizon 2020 programme (grant agreement No 731060) (JLF), EMBL Interdisciplinary Postdoc Programme under Marie Curie COFUND actions MSCA-COFUND-FP (grant agreement number: 847543) (MS), German Center for Infection Research, DZIF (FH), DFG-FR 2140/10-1 (AMB), DFG research networks SFB 1129, SPP 2332 and grant FR 2140/10-1 (FF), CSSB KIF-002 (TWG and KG), DFG research networks SPP 2225 (TWG), Wellcome Trust grants 107806/Z/15/Z and 209250/Z/17/ Z (KG), BMBF grant 05K18BHA (KG), DFG INST 152/ 772-1, 774-1, 775-1, 777-1 FUGG (CSSB cryoEM facility).

## Author contributions

JLF and EP cultured blood stage parasites, AMB and FH prepared mosquito stage parasites. AMB generated transgenic *P. berghei* lines and performed fluorescence localisation experiments. JLF performed fluorescence microscopy, EM grid preparation, FIB-milling and electron microscopy data collection. JLF and VP performed tomography data processing, subvolume averaging and polarity analysis. VP and DV performed additional analysis on tomography data including ellipticity analysis and fitting. VP and MS conducted distance and angle analysis in tomography data. JLF, VP and MS performed tomogram segmentation. JLF, VP, AMB and EP prepared figures. JK, FF, TWG and KG supervised the project. JLF and VP wrote the original manuscript text and it was reviewed and edited by JLF, VP, DV, MS, FH, EP, JK, FF, TWG, KG.

## Funding

## Competing interests

The authors declare no competing interests.

## Ethics statement

For in vivo experiments including parasite propagation and mosquito infections, 8–10-week-old female Swiss mice obtained from Janvier labs were used. All animal experiments were performed according to European regulations concerning FELASA category B and GVSOLAS standard guidelines. Animal experiments were approved by German authorities (Regierungspraesidium Karlsruhe, Germany), § 8 Abs. 1 Tierschutzgesetz (TierSchG) under the license G-111/20 and were performed according to National and European regulations.
