## [Peer Review File · Nature Communications]

Reviewer comments, first round –

Reviewer #1 (Remarks to the Author):

Variable microtubule architecture in the malaria parasite
Ferreira, Grünewald, et al.

The authors present a study on microtubule architecture and protofilament composition in the malaria parasite *Plasmodium falciparum*. They used in situ cryo-electron tomography and subsequent sub volume averaging to analyze the changing microtubule cytoskeleton of these organisms, a rare occasion where microtubules deviate from the canonical 13-protofilament structure. While different microtubule protofilament numbers have been seen in other organisms (e.g., some types of insect sperms), the authors claim the variability seen here is quite unique. Of particular interest is the different degrees of microtubule variability within life cycle stages of plasmodium.

The work is technically sound and of high quality, especially the raw tomographic imaging, which a key prerequisite for any further computational analysis and statistics. 3-D Image analysis revealed an interesting variation of microtubule architecture, protofilament composition and inner structures from associated proteins. They authors present a vast amount of data and 3-D analysis of these microtubules in various stages of a plasmodium life cycle. Overall, though being mostly descriptive, this is an exciting paper and should be published.

A few minor comments:

In plasmodium, "squeezed" microtubules have been seen before by vitrified sectioning, but it was mostly attributed to compression from the microtome sectioning through ice. Maybe this was a real feature after all ...

To my knowledge, microtubules vary protofilament numbers according to the composition of the solution they are in, at least in vitro. Do the authors have any suspicion to what these different sizes may trigger, other than MAPs and a lack of nucleation?

Again, in my experience, larger tubes tend to squeeze easier than tight ones. Could this opposing observation that was made here be mostly attributed to MAP association, but maybe different ones from the inner decoration?

If I understand this correct, the ILH is not only controlling the overall supertwist in a cell, but also "squeezing" the microtubule into an oval form? I don't think any such inner-MT protein decoration in higher eukaryotes ever showed such a regular pattern.

In vitro microtubules assembled in the presence of DMSO, or Glycerol form about 10% or less 15 and 16 MTs. Without these agents, 13 protofilament MTs are the dominant form, not 14. 15-protofilament microtubules and larger are typically true helical structures (i.e., no seam).

The acronym "APR" should be defined somewhere.

The sentence at line 527 (instead ...) seems incomplete.

Line 530: I don't understand the following statement: this it is also difficult to reconcile the supertwist of non-canonical microtubules with a linker having a preference for a specific protofilament. Why should this be true (e.g., see kinesins)?

The outlook is a bit overly optimistic ... However, the paper shows a vast amount of new data on microtubules, which at this point are mostly descriptive, but could trigger new approaches towards a better understanding of plasmodium, and possibly Malaria in a bigger picture.

Reviewer #2 (Remarks to the Author):

Plasmodium falciparum is the leading cause of severe malaria in humans and is a leading cause of morbidity and mortality globally. *Plasmodium* complex lifecycle alternates between the mosquito and human host, with each stage possessing diverse microtubule cytoskeletal structures essential for its proliferation, growth, or transmission. Fundamental understandings of the structural organization and molecular composition of the cytoskeletal structures at the different lifecycles of the parasite are likely to underlie new contributions to *Plasmodium* cell biology and the development of antimalarial strategies.

Significant progress has been made in uncovering structural features of *Plasmodium* parasites and other apicomplexan parasites using cryo-ET. However, sample thickness imposed a limit on the type of samples that could be imaged. Using cryo-focused ion beam (cryo-FIB) milling, the authors generated thin lamella of four parasite life stages and investigated the different types of microtubules using cryo-ET. They find structural diversity in the microtubules across the parasite's life stages, with an ILH observed only in the mosquito stage, and a mixture of singlets, doublets, triplets, and quadruplets with a distributed number of protofilaments found only in gametocytes. The work is well performed, and the structural information will nevertheless be useful for further refinement; however, some of the statements made by the authors are missing the data to support them.

Major comments:

There is no sufficient data to support the authors' conclusion that the ILH of the sporozoites SPMTs compose of 10 copies of TrxL1. The resolution of the SPMTs density map is insufficient to determine a reliable fit of the TrxL1 model. The authors suggested fitting is based on the assumption that TrxL1 localizes to the SPMTs with the same organization that was determined for *Toxoplasma* tachyzoites. In addition, the authors fitted the model as half arcs and not as a whole (due to differences in ellipticity). The authors should provide evidence for the localization of TrxL1 in the *Plasmodium* sporozoites microtubules (antibody labeling/ tag/ purified microtubules proteomics).

Specific comments:

In Figure 2C, the authors only present the fitting of TrxL1 and SPM1 and are missing the tubulin fitting. Also, it needs to be clarified if the fitting is based on the 7MIZ model or the structural prediction of either *Plasmodium* or *Toxoplasma* TrxL1.

Please comment if the polarity of the microtubules (that was measured independently) agrees with the 7MIZ model fitting.

Please elaborate on how the specific fitting of α/β units into the density map was determined.

The authors should include unannotated images in their figures. It is extremely difficult to judge the authors' interpretation of specific densities in the tomogram slices as the colored annotations completely obscure them. In addition, some of the authors' interpretations are based on 3D information, and the tomograms that support these observations must be included as supplementary data. It is impossible to comment on the authors' description of the ookinete conoid or apical collar just based on their annotation.

The authors discuss the different stages of gametocytes they imaged, but the results do not address their similarities or differences. Including representative tomograms or at least images of the different stages in the supplementary material will be nice.

Minor comments:

The order of supplemental figures does not correspond to their order in the main text.

Ln 103: "In the invasive forms of apicomplexan parasites, a diverse MTOC at their apical end, the Apical Polar Rings (APR), coordinates the nucleation and higher order spatial control of SPMTs, but

the specific mechanism remains unknown.”

Since the nucleation mechanism is unknown, there is no evidence to say that the MTOC coordinates the SPMTs nucleation, and the authors should only point to its role in spatial organization.

Ln 169: which in the Toxoplasma ILH, fills in one of the two “interruptions”.

In Toxoplasma’s ILH, TrxL2 is present in two positions, between position 12 and 13 and between position 6 and 7; therefore, while it may completely fill one of the “interruptions”, it is also part of the second “interruption”.

Ln 208: Please add statistics to the ookinete conoid measurements, or if N=1, please clarify that in the text.

Ln 275: Correct to Fig. 5C

Ln 293: Correct to Fig. 5B

Ln 333: “In two tomograms we could see the full sporozoite APR with a set of regularly interspaced 13+1 SPMTs”

Please clarify 13+1. If the number is different than expected, please include a reference.

Ln 413 and Ln 447: Figure 6 and 7B- the authors speculate that the radial polarity of the merozoites SPMTs is random, but it is not sufficient to claim that the ILH determines polarity.

Ln 506: “The TrxL1 expression profile in published data is consistent with our observation”
Please refer to the proteomics data that supports this statement (available on PlasmoDB). Also, what about the expression levels/proteomics data on SPM1?

Ln 526: “This was unanticipated as the seam is the only asymmetric feature that can be accessed by externally binding MAPs”

The orientation of the seam opposite to the IMC was shown in Toxoplasma tachyzoites SPMTs in Sun et al., PNAS 2022.

Ln 720: Please specify the published parameters used and add a reference.

Fig. S2A: Please clarify what data was used to determine the protofilament angle measurement for Toxoplasma and include the number of measurements and appropriate reference to the original data.

Fig. S6E: Please clarify the “alveolin layer” is this the same as the apical collar?

Reviewer #3 (Remarks to the Author):

The authors have examined the microtubule cytoskeleton of Plasmodium falciparum at different stages of the lifecycle, using in situ electron cryotomography and subtomogram averaging. Live parasites were vitrified on EM grids before generating lamella by FIB milling and then multiple tomograms were acquired by TEM. Subvolume averaging was used to analyse individual microtubules.

This a technically impressive study.

1) The study reveals the ookinete SPMTs as structures with 13-protofilaments with twice-interrupted luminal helices that interact with the apical polar ring. The authors interpret the extra EM density in the P. falciparum ILH as being occupied by 10 copies of PfTrxL1. In the absence of any biochemical data (and the lack of a gene encoding PfTrxL2), the authors may wish to be a little more cautious in this interpretation.

2) The study confirms the presence of a conoid-like structure at the apex of the ookinete. The data

illustrate the two apical collar layers between the SPMTs and IMC. The SPMTs are positioned well below the conoid. The authors may wish to comment on whether the conoid might play a role in initiating the SPMTs in this stage. Interestingly the SPMT presented in Fig S3A appears capped. The authors could comment.

3) Line 240. The authors state: "However, in contrast to the SPMT, where minus ends lacked a capping density, spindle microtubules had a clear cap consistent with γ TuRC (Fig. S3C)." The authors should state the number of spindle microtubules that were observed to have this cap? What verification is available that the cap represents γ TuRC? Is there related literature to support the suggestion that spindle microtubule ends are capped by γ TuRC in plasmodium schizonts? Or do the authors have anti- γ tubulin immunofluorescence microscopy data to support this conclusion?

4) Similar questions relate to the suggestion that gametocyte nuclear microtubules are capped with γ TuRC.

5) In developing gametocytes, the authors report the presence of unusual large microtubules (27-37 nm/ 13-18 protomers). This result is surprising. Previous studies have reported the diameter of SPMTs in gametocytes as 27 ± 4 nm (3) (2). An early study reported microtubules depolymerisation at low temperature, resulting in short segments of 31-34 nm diameter tubules (4). For the current work the authors resuspend isolated gametocytes in prewarmed RPMI without serum. However, the previous study (3) resuspended gametocytes in complete culture medium and incubated at 37°C for 2 h to allow recovery of microtubules. The authors should comment on whether the difference in protocol might be responsible for the different observations.

6) Page 20 Line 379. The authors state: "Thus, it is likely that there is a unique protein, linking the SPMTs to the IMC in all parasite forms analysed." The data are not sufficient to make such a strong statement.

7) The authors provide a detailed discussion of the implications of the findings. An additional point that would be worth discussing is the recent finding that SPMTs are polyglutamated (2) (1), which may contribute to the stability of SPMTs in the absence of (the presumed) γ TuRC stabilisation. The authors could also discuss a recent study that suggested that the first set of SPMTs in gametocytes may be nucleated from the cytoplasmic surface of the MTOC (2).

Minor comments.

Figure 1C. The antibodies used for fluorescence labelling should be pointed out in the figures and figure legends.

Line 275. "Although some canonical microtubules were present, these only accounted for 9% of the population, while the majority of singlet microtubules had 17-protofilaments (40%) (Fig. 3C)". Fig. 3C should be Fig. 5C.

"Nuclear microtubules were seen frequently in stage III/IV parasites, and a full spindle pole body was present in one tomogram (Fig. 5C)." Fig. 5C should be Fig. 5B.

There are several instances in which the order of presentation of the data does not match the order of presentation of the figures. This should be corrected (where possible).

1. Bertiaux E, Balestra AC, Bournonville L, Louvel V, Maco B, et al. 2021. Expansion microscopy provides new insights into the cytoskeleton of malaria parasites including the conservation of a conoid. *PLoS biology* 19:e3001020
2. Dearnley MK, Yeoman JA, Hanssen E, Kenny S, Turnbull L, et al. 2012. Origin, composition, organization and function of the inner membrane complex of *Plasmodium falciparum* gametocytes. *J Cell Sci* 125:2053-63
3. Li J, Shami GJ, Cho E, Liu B, Hanssen E, et al. 2022. Repurposing the mitotic machinery to drive

cellular elongation and chromatin reorganisation in *Plasmodium falciparum* gametocytes. *Nat Commun* 13:5054

4. Tilney LG, Porter KR. 1967. Studies on the microtubules in heliozoa: II. The effect of low temperature on these structures in the formation and maintenance of the axopodia. *The Journal of cell biology* 34:327

Reviewer #1 (Remarks to the Author):

*Variable microtubule architecture in the malaria parasite
Ferreira, Grünewald, et al.*

The authors present a study on microtubule architecture and protofilament composition in the malaria parasite Plasmodium falciparum. They used in situ cryo-electron tomography and subsequent sub volume averaging to analyze the changing microtubule cytoskeleton of these organisms, a rare occasion where microtubules deviate from the canonical 13-protofilament structure. While different microtubule protofilament numbers have been seen in other organisms (e.g., some types of insect sperms), the authors claim the variability seen here is quite unique. Of particular interest is the different degrees of microtubule variability within life cycle stages of plasmodium.

The work is technically sound and of high quality, especially the raw tomographic imaging, which a key prerequisite for any further computational analysis and statistics. 3-D Image analysis revealed an interesting variation of microtubule architecture, protofilament composition and inner structures from associated proteins. They authors present a vast amount of data and 3-D analysis of these microtubules in various stages of a plasmodium life cycle. Overall, though being mostly descriptive, this is an exciting paper and should be published.

Many thanks to the reviewer for the time and effort put into the review of this manuscript and thank you for agreeing that our paper is exciting and should be published.

A few minor comments:

In plasmodium, “squeezed” microtubules have been seen before by vitrified sectioning, but it was mostly attributed to compression from the microtome sectioning through ice. Maybe this was a real feature after all ...

Yes, we observed that in mosquito forms the microtubules are elliptical (page 9, line 183) and we also see that the giant microtubules in gametocytes are not always perfectly round (seen in for example new supplemental movie S4).

To my knowledge, microtubules vary protofilament numbers according to the composition of the solution they are in, at least in vitro. Do the authors have any suspicion to what these different sizes may trigger, other than MAPs and a lack of nucleation?

The question of how the diverse sizes of microtubules are nucleated is one of the most interesting ones, which we would like to pursue in the future. Another reason, other than those mentioned

by the reviewer, for the consistently larger tubule diameter could be the expression of alpha-tubulin-II in sexual stages, as stated in the discussion (page 25, paragraph from line 476).

Again, in my experience, larger tubes tend to squeeze easier than tight ones. Could this opposing observation that was made here be mostly attributed to MAP association, but maybe different ones from the inner decoration?

We agree that larger tubules are likely easier to deform laterally (made elliptical - and we do see this, as mentioned above), but the larger tubules are expected to be more resistant to bending (now clarified in the text: page 26, line 498). Whether this observation can be extended beyond the relatively well-studied 15 protofilament microtubules to even larger tubules remains to be determined and therefore we cannot disentangle the role of protofilament number and potential additional MAPs at this stage.

“We expect that giant microtubules will be significantly stiffer (less likely to bend) than the canonical form.”

If I understand this correct, the ILH is not only controlling the overall supertwist in a cell, but also “squeezing” the microtubule into an oval form? I don’t think any such inner-MT protein decoration in higher eukaryotes ever showed such a regular pattern.

Yes, this is correct, and we see that *Plasmodium* microtubules are slightly more elliptical than those of *Toxoplasma*. Whether this is due to the difference, e.g., in the N-terminal helix sequences between the two organisms or due to detergent solubilisation that was used in the *T. gondii* studies remains to be seen. This structural difference is shown in supplemental figure S1e, and is now added to the manuscript results section to highlight this in the text (page 9, line 189).

“Comparisons of the predicted structure of Plasmodium’s PfTrxL1 with that of TgTrxL1 from T. gondii, showed that the largest difference is at this N-terminal helix (Fig. S1e), responsible for a large part of the subunit-subunit interface, and may play a role in the increased ellipticity.”

In vitro microtubules assembled in the presence of DMSO, or Glycerol form about 10% or less 15 and 16 MTs. Without these agents, 13 protofilament MTs are the dominant form, not 14. 15-protofilament microtubules and larger are typically true helical structures (i.e., no seam).

We agree that depending on the *in vitro* conditions used, the distribution of protofilament numbers is different and this also depends on the species studied. To make sure we are more accurate in our introduction, we have changed the wording (page 4, line 93) to:

“Microtubules assembled in vitro, from purified tubulin, form 9- to 16-protofilament microtubules, with different distributions of protofilament numbers depending on the species and conditions used^{1,16}.”

The acronym “APR” should be defined somewhere.

“APR” has been defined in the introduction (page 5, line 103).

The sentence at line 527 (instead ...) seems incomplete.

Thanks for noticing, this is now corrected.

Line 530: I don't understand the following statement: this it is also difficult to reconcile the supertwist of non-canonical microtubules with a linker having a preference for a specific protofilament. Why should this be true (e.g., see kinesins)?

The uniform orientation of ILH-containing SPMTs relative to the IMC suggest that there must be (one or several) “special” protofilaments to which the IMC-linker protein binds preferentially (directly or indirectly). This statement was meant to reinforce that it is unlikely that there are “special” protofilaments in the larger, “super-twisted” tubules, as there are micrometer-sized patches where any single protofilament is oriented opposite to the IMC. We have now clarified this in the text (page 28, line 543).

“...it is also difficult to reconcile the supertwist of non-canonical microtubules with a linker having a preference for a specific protofilament (as different protofilaments will be directed at the IMC as the microtubule twists).”

The outlook is a bit overly optimistic ... However, the paper shows a vast amount of new data on microtubules, which at this point are mostly descriptive, but could trigger new approaches towards a better understanding of plasmodium, and possibly Malaria in a bigger picture.

We wanted the outlook to be a thought-provoking section, aiming to reinforce the fact that we know relatively little about the cytoskeleton of less extensively studied clades. Prior to undertaking this study some of us would be ready to assume that most eukaryotes would have “relatively similar” cytoskeletal structures. While this may still be the case, we hoped to stimulate researchers to examine the microtubules in their tomographic data when they come across them.

Reviewer #2 (Remarks to the Author):

Plasmodium falciparum is the leading cause of severe malaria in humans and is a leading cause of morbidity and mortality globally. Plasmodium complex lifecycle alternates between the mosquito and human host, with each stage possessing diverse microtubule cytoskeletal structures essential for its proliferation, growth, or transmission. Fundamental understandings of the structural organization and molecular composition of the cytoskeletal structures at the different lifecycles of the parasite are likely to underlie new contributions to Plasmodium cell biology and the development of antimalarial strategies.

Significant progress has been made in uncovering structural features of Plasmodium parasites and other apicomplexan parasites using cryo-ET. However, sample thickness imposed a limit on the type of samples that could be imaged. Using cryo-focused ion beam (cryo-FIB) milling, the authors generated thin lamella of four parasite life stages and investigated the different types of microtubules using cryo-ET. They find structural diversity in the microtubules across the parasite's life stages, with an ILH observed only in the mosquito stage, and a mixture of singlets, doublets, triplets, and quadruplets with a distributed number of protofilaments found only in gametocytes. The work is well performed, and the structural information will nevertheless be useful for further refinement; however, some of the statements made by the authors are missing the data to support them.

We thank the reviewer for their time, constructive feedback and recognizing the importance of our work. We believe that with the changes we have made to the manuscript and by adding new data and supplemental figures, the data now supports our statements more concretely.

Major comments:

There is no sufficient data to support the authors' conclusion that the ILH of the sporozoites SPMTs compose of 10 copies of TrxL1. The resolution of the SPMTs density map is insufficient to determine a reliable fit of the TrxL1 model. The authors suggested fitting is based on the assumption that TrxL1 localizes to the SPMTs with the same organization that was determined for Toxoplasma tachyzoites. In addition, the authors fitted the model as half arcs and not as a whole (due to differences in ellipticity). The authors should provide evidence for the localization of TrxL1 in the Plasmodium sporozoites microtubules (antibody labeling/ tag/ purified microtubules proteomics).

This is correct, at the resolution we achieved (~ 2 nm), it is not possible to unambiguously determine whether the density we observe in *P. falciparum* and *P. berghei* corresponds specifically to PfTrxL1/PbTrxL1. However, we reasoned by analogy, given the high degree of sequence conservation, the structural similarity between TgTrxL1 and the PfTrxL1 structure prediction and subjectively, the good fit of the atomic model in the *P. falciparum* density map

that it is very likely that both PfTrxL1 and PfSPM1 are present in the ILH. The reason for rigid body fitting the half arcs separately (as indicated in the method section: page 38, line 782) is the difference in ellipticity between the two density maps.

We have changed the wording of this sentence from “*We are therefore confident that P. falciparum ILH consists of 10 copies of PfTrxL1, likely with an equivalent number of PfSPM1 separated into two half-crescents (Fig. 2c).*” to “*We therefore suggest that...*”.

Furthermore, we have now added a new supplemental figure (Fig. S2) providing new data in which we endogenously tagged PbSPM1 and PbTrxL1 with GFP. In this new figure, we now show that the localisation of TrxL1-GFP and SPM1-GFPs in sporozoites is consistent with SPMTs, as has been demonstrated previously (in for example: Spreng, B. *et al. Microtubule number and length determine cellular shape and function in Plasmodium. EMBO J.* **38**, e100984 (2019)). A new methods section has been added for generation of tagged lines and reference to this new figure has been made in the text (page 9, line 173).

Our rationale for the ILH being made up of SPM1 and TrxL1 in *Plasmodium* is now strong and we outline it in the following paragraph (page 8, line 165):

“The T. gondii ILH consists of thioredoxin-like proteins 1 and 2 (TrxL1, TrxL2) and subpellicular microtubule protein 1 (SPM1). We hypothesised that the ILH in Plasmodium and Toxoplasma consist of homologous proteins: Firstly, Plasmodium spp have homologues of TrxL1 (PfTrxL1) and SPM1 (PfSPM1) but lack TrxL2, which in the Toxoplasma ILH, is present in both ILH “interruptions” and completely fills one of the two “interruptions”. Consistently, we exclusively observed twice-interrupted luminal helices in Plasmodium. Secondly, compatible with the ILH being made up of SPM1 and TrxL1 in Plasmodium, we saw high expression of PbSPM1-GFP and PbTrxL1-GFP in our endogenously-tagged sporozoite lines (Fig. S2), with localisations consistent with sporozoite SPMTs (as shown in²⁸). And finally, PfTrxL1 and TgTrxL1 and, PfSPM1 and TgSPM1 are highly conserved¹² (Fig S1a,b), and the model of T. gondii SPMT assembly (pdb 7MIZ) fits well into our EM maps (Fig. 2c, d). Due to the asymmetric nature of the ILH, there is only one unique way that the ILH structure from T. gondii can be fitted into the EM density of Plasmodium microtubules. We therefore suggest that P. falciparum ILH consists of 10 copies of PfTrxL1, likely with an equivalent number of PfSPM1 separated into two half-crescents (Fig. 2c).”

There is no other conclusive experiment that could give us higher confidence in TrxL1 copy number and localisation other than performing single particle cryoEM on detergent solubilised parasites. However, as has been demonstrated, this would merit an independent publication. Antibody labeling could be performed via the Tokuyashu method, however, this method is notoriously unreliable in terms of precision labelling and antibody labeling itself introduces a

large “circle of confusion” due to the random antibody orientations. Similarly, fluorescence-based (super-resolution) labeling can only show rough localisation. Purifying *Plasmodium* microtubules is far from a trivial task, as has been demonstrated recently (Hirst, W. G. *et al. Purification of functional Plasmodium falciparum tubulin allows for the identification of parasite-specific microtubule inhibitors. Curr. Biol.* **32**, 919-926.e6 (2022)) with schizonts, which can be grown to much higher densities than mosquito forms.

As the main aim of the paper is to convey the diversity of microtubule structures within *Plasmodium* and not overly focus on the exact position, structures and orientation of the ILH proteins, we strongly believe that by adding the new PbTrxL1-GFP and PbSPM1-GFP expression and localisation data, and clarifying the text, as above, our rationale for this section has been strengthened and is now suitable for publication.

Specific comments:

In Figure 2C, the authors only present the fitting of TrxL1 and SPM1 and are missing the tubulin fitting.

This is correct, tubulin fitting is shown in Fig 2d , it was omitted in panel c for clarity. The difference in protofilament orientation, determined by rigid body fitting, is shown quantitatively in figure S1. There we choose to represent protofilaments schematically, as overlaying two sets of atomic models would be counter-productive.

We have now clarified the legend to Fig S1 accordingly:

*c. Angle relative to protofilament $n+1$. Dotted line denotes the inter-protofilament angle of a theoretical 13 protofilament microtubule. This was determined by fitting individual protofilaments of *T. gondii* tubulin into the *P. falciparum* density and measuring the angle relative to *pdb 7MIZ*. **d.** Both the *Toxoplasma gondii* (red) structure and our mosquito form *Plasmodium* (blue) structures have elliptical cross-sections. The average ellipticity (determined using the same measurements as in panel A) of a *Plasmodium* (blue ellipse) and *T. gondii* (red ellipse) microtubule and the relative tilt between protofilaments (blue and red bars) are superimposed onto a mean projection of the *Plasmodium* EM map.*

*Also, it needs to be clarified if the fitting is based on the 7MIZ model or the structural prediction of either *Plasmodium* or *Toxoplasma TrxL1*.*

This is stated in the text: “...the model of *T. gondii* SPMT assembly (*pdb 7MIZ*) fits well into our EM maps (Fig. 2c, d).

We have now modified the figure legend to refer to TgTrxL1 and TgSPM1 rather than simply TrxL1 and SPM1 (page 11, line 215):

“c. Isosurface representation of the EM map with TgTrxL1 and TgSPM1 fitted into the EM density. p1-p13 = protofilament numbers, dotted line = seam position.”

Please comment if the polarity of the microtubules (that was measured independently) agrees with the 7MIZ model fitting.

Microtubule polarity can be unambiguously determined from the protofilament tilt (shown in Fig S5d), however we did not credit the authors that first published this observation (Sosa and Chrétien, 1998), this is now corrected (page 6, line 131).

This was taken into account while fitting 7MIZ and is shown quantitatively (by comparing protofilament tilt in Fig S1).

Please elaborate on how the specific fitting of α/β units into the density map was determined.

The ILH striation fully defines the position of α/β subunits within the density. Clearly, this relies on PfSPM1 making the same contacts with tubulin and PfTrxL1 as those observed in *T. gondii*. We believe this is a reasonable assumption given the high degree of sequence conservation. The uncertainty regarding whether or not *T. gondii*, *P. falciparum* and *P. berghei* have identical architectures is sufficiently expressed, with clarification suggested by the reviewers, in our fitting section (page 9, line 176).

*“Due to the asymmetric nature of the ILH, there is only one unique way that the ILH structure from *T. gondii* can be fitted into the EM density of Plasmodium microtubules. We therefore suggest that *P. falciparum* ILH consists of 10 copies of PfTrxL1, likely with an equivalent number of PfSPM1 separated into two half-crescents (Fig. 2c). This also provided us with an unambiguous method of identifying both the seam position and alpha-beta tubulin location in our structure (Fig 2c,d).”*

The authors should include unannotated images in their figures. It is extremely difficult to judge the authors' interpretation of specific densities in the tomogram slices as the colored annotations completely obscure them. In addition, some of the authors' interpretations are based on 3D information, and the tomograms that support these observations must be included as supplementary data. It is impossible to comment on the authors' description of the ookinete conoid or apical collar just based on their annotation.

Slices through all tomograms that are segmented are now included as a new supplemental figure (S3) and movies for all four have been added as supplemental data (supplemental movies S1, S2, S3, S4). References to these have been added to the figure legends for example in Fig: 2 legend:

“f. Segmented sporozoite apical pole. The tubulin density (13 protofilaments) of two SPMTs was hidden to reveal the ILH. Unsegmented slice through tomogram shown in Fig. S3a and supplemental movie S1.”

The authors discuss the different stages of gametocytes they imaged, but the results do not address their similarities or differences. Including representative tomograms or at least images of the different stages in the supplementary material will be nice.

As this manuscript focused on changes in the microtubule structure, and we saw no difference in microtubule structure (protofilament number distribution) between different gametocyte stages, we decided that adding more data on other morphological differences (which have been characterised elsewhere) would complicate the manuscript and would be beyond the scope.

Minor comments:

The order of supplemental figures does not correspond to their order in the main text.

We have now corrected this.

Ln 103: “In the invasive forms of apicomplexan parasites, a diverse MTOC at their apical end, the Apical Polar Rings (APR), coordinates the nucleation and higher order spatial control of SPMTs, but the specific mechanism remains unknown.”

Since the nucleation mechanism is unknown, there is no evidence to say that the MTOC coordinates the SPMTs nucleation, and the authors should only point to its role in spatial organization.

Thank you for noticing this inconsistency, we have now changed it to (page 5, line 101):

“In the invasive forms of apicomplexan parasites, a diverse MTOC at their apical end, the Apical Polar Rings (APR), coordinates the higher order spatial control of SPMTs, but the specific mechanism remains unknown.”

Ln 169: which in the Toxoplasma ILH, fills in one of the two “interruptions”.

In Toxoplasma's ILH, TrxL2 is present in two positions, between position 12 and 13 and between position 6 and 7; therefore, while it may completely fill one of the "interruptions", it is also part of the second "interruption".

Thank you for pointing this out, we have now clarified this in the text and changed it to (page 8, line 169):

"Toxoplasma ILH, is present in both "interruptions" and completely fills one of the two "interruptions"."

Ln 208: Please add statistics to the ookinete conoid measurements, or if N=1, please clarify that in the text.

We observed the conoid in one lamella and clarified this in the text to (page 11, line 215):

"At the apical end of the ookinete, we observed a structure consistent with a classical conoid made up of a unique tubulin structure (N=1, Fig. 3e)."

Ln 275: Correct to Fig. 5C

Done, thanks.

Ln 293: Correct to Fig. 5B

Done, thanks.

Ln 333: "In two tomograms we could see the full sporozoite APR with a set of regularly interspaced 13+1 SPMTs"

Please clarify 13+1. If the number is different than expected, please include a reference.

Sporozoites have a cluster of closely spaced microtubules and then one lone microtubule on the opposite side of the cell body. In each *Plasmodium* species the number of clustered microtubules is different. For example, *P. berghei* has 15(clustered)+1(lone). We have now clarified this in the text (page 18, line 346).

"In two tomograms we could see a full sporozoite APR with the complete set of 13 closely spaced SPMTs opposed by a single SPMT (Fig. 2e, f, Fig. S6)."

Ln 413 and Ln 447: Figure 6 and 7B- the authors speculate that the radial polarity of the merozoites SPMTs is random, but it is not sufficient to claim that the ILH determines polarity.

This is correct, ILH correlates with radial polarity but this is insufficient to claim causation. We have changed the following text accordingly (page 27, line 540):

“Instead, it is likely the ILH that sets the rotational orientations.”

now reads

“Instead, the ILH could be setting the rotational orientation directly.”

Ln 506: “The TrxL1 expression profile in published data is consistent with our observation” Please refer to the proteomics data that supports this statement (available on PlasmoDB). Also, what about the expression levels/proteomics data on SPM1?

Thank you - we have changed this section to now include the references for transcriptomics data as well as reference to the new data we have added to the manuscript, where we endogenously-tagged pbSPM1 and pbTrxL1 with GFP and show their expression and SPMT localisation in sporozoites (new Figure S2). This now reads (page 26, line 516):

“The TrxL1 expression profile in published data is consistent with our observation, with its transcription strongly upregulated in ookinetes and sporozoites, and very little relative expression in the blood states^{38,47}. In our TrxL1-GFP and SPM1-GFP lines, we saw high levels of expression in sporozoites at localisations consistent with SPMTs (Fig S2)”

Ln 526: “This was unanticipated as the seam is the only asymmetric feature that can be accessed by externally binding MAPs” The orientation of the seam opposite to the IMC was shown in Toxoplasma tachyzoites SPMTs in Sun et al., PNAS 2022.

We have adapted the text and this sentence now reads (page 27, line 538):

“This is consistent with and clarifies previous data on seam orientation from flattened detergent solubilised T. gondii¹⁴, but remains surprising as the seam is the only asymmetric feature that can be accessed by externally binding MAPs.”

Ln 720: Please specify the published parameters used and add a reference.

The published parameters were listed in Table S2. We have now added a reference to this and a reference to original source, as below (page 35, line 730):

“Helical parameters were measured directly using class averages with the exception of 13- and 15-protofilament classes where tubulin subunits were not resolved and published parameters were used (table S2)⁷⁰.”

Fig. S2A: Please clarify what data was used to determine the protofilament angle measurement for Toxoplasma and include the number of measurements and appropriate reference to the original data.

We have included this among modifications listed above.

Fig. S6E: Please clarify the “alveolin layer” is this the same as the apical collar?

We were referring to the “hairy” layer on the inner leaflet of the inner IMC membrane. We have removed this label from the figure as it was a speculation.

Reviewer #3 (Remarks to the Author):

The authors have examined the microtubule cytoskeleton of Plasmodium falciparum at different stages of the lifecycle, using in situ electron cryotomography and subtomogram averaging. Live parasites were vitrified on EM grids before generating lamella by FIB milling and then multiple tomograms were acquired by TEM. Subvolume averaging was used to analyse individual microtubules.

This a technically impressive study.

We thank the reviewer for their thoughtful comments and suggestions.

1) The study reveals the ookinete SPMTs as structures with 13-protofilaments with twice-interrupted luminal helices that interact with the apical polar ring. The authors interpret the extra EM density in the P. falciparum ILH as being occupied by 10 copies of PfTrxL1. In the absence of any biochemical data (and the lack of a gene encoding PfTrxL2), the authors may wish to be a little more cautious in this interpretation.

We agree and have now modified this section (page 9, line 178).

We have changed the wording of this sentence from “*We are therefore confident that P. falciparum ILH consists of 10 copies of PfTrxL1, likely with an equivalent number of PfSPM1 separated into two half-crescents (Fig. 2c).*” to “*We therefore suggest that...*”.

Furthermore, we have now added a new supplemental figure Fig. S2 with new data in which we endogenously-tagged PbSPM1 and PbTrxL1 with GFP and show their expression in sporozoites with clear localisation consistent with sporozoite SPMTs. Reference to this new figure has been made in the text (page 9, line 171):

“Secondly, compatible with the ILH being made up of SPM1 and TrxL1 in Plasmodium, we saw high expression of PbSPM1-GFP and PbTrxL1-GFP in our endogenously-tagged sporozoite lines (Fig. S2), with localisations consistent with sporozoite SPMTs (peripheral staining of half to two-thirds cell length; as shown in²⁸).”

2) The study confirms the presence of a conoid-like structure at the apex of the ookinete. The data illustrate the two apical collar layers between the SPMTs and IMC. The SPMTs are positioned well below the conoid. The authors may wish to comment on whether the conoid might play a role in initiating the SPMTs in this stage.

Thank you for this comment, yes the conoid appears to almost be “floating” far away from the APRs or SPMTs. As we don’t have sufficient data on the conoid (one reasonably thick tomogram), we would prefer not to speculate on this at this stage.

Interestingly the SPMT presented in Fig S3A appears capped. The authors could comment.

This is interesting and the “cap” looks distinct from what we see at the spindle minus ends.

We have commented on this in the text (page 18, line 338):

“In all invasive forms, their minus ends were blunt and lacked a γ TuRC cap, although there may be additional microtubule associated proteins (MAPs) in the lumen of ookinete minus ends (Fig S3a).”

3) Line 240. The authors state: “However, in contrast to the SPMT, where minus ends lacked a capping density, spindle microtubules had a clear cap consistent with γ TuRC (Fig. S3C).” The authors should state the number of spindle microtubules that were observed to have this cap? What verification is available that the cap represents γ TuRC? Is there related literature to supports the suggestion that spindle microtubule ends are capped by γ TuRC in plasmodium schizonts? Or do the authors have anti- γ tubulin immunofluorescence microscopy data to support this conclusion?

All spindle microtubule minus ends that we could confidently assign as minus ends (fully within the lamella) in our tomograms had a clear capping density. Due to the random placement and

thickness of lamella, it is rare to catch the spindle body, but when we did, we could see clear capping densities. The number of minus-ends is indicated by the number of averaged particles indicated in Fig. S4.

We do not know for certain that this is γ TuRC but rather wanted to highlight to other researchers that there is a clear capping density (and minus ends look distinct from SPMT minus ends) and suggest that this could be γ TuRC. γ -tubulin has been associated with spindle minus-ends in *Plasmodium* schizonts (Simon CS. et al (2021) *Unconventional centrosome organization in malaria parasites*. Life Science Alliance. DOI: 10.26508/lsa.202101199) - this is now adequately referenced in the text and we have changed our text to make it clear that this is a suggestion.

To verge on the side of caution, we have also modified the relevant sentence (page 13, line 249):

“However, in contrast to SPMTs which were uncapped, spindle microtubules contained a capping density at their minus ends which resembled γ TuRC (Fig. S4c)³¹. “

4) Similar questions relate to the suggestion that gametocyte nuclear microtubules are capped with γ TuRC.

As above with schizonts, we have clarified the text for gametocytes in the results and discussion (page 16, line 305).

“This was unexpected as, in all minus ends that we observed, were - irrespective of the microtubule diameter and protofilament number - clearly capped with a structure which could be γ TuRC (although it is flattened relative to the canonical form), which has, so far, been exclusively observed with canonical microtubules (Fig. S4d).”

5) In developing gametocytes, the authors report the presence of unusual large microtubules (27-37 nm/ 13-18 protofilaments). This result is surprising. Previous studies have reported the diameter of SPMTs in gametocytes as 27 ± 4 nm (3)

Yes, we were surprised too! But this is the first study to accurately measure structures (rather than individual microtubules) solved *in vivo* at native (vitrified, unfixed and unstained) conditions. Many different reasons (including shrinkage due to resin embedding, heavy metal staining and sectioning artifacts) could result in the discrepancy in the measurements. Interestingly, doublet microtubules are clearly seen in some of this data (eg: REF 3 below) but the authors do not mention or comment on them.

(2). An early study reported microtubules depolymerisation at low temperature, resulting in short segments of 31-34 nm diameter tubules (4). For the current work the authors resuspend isolated gametocytes in prewarmed RPMI without serum. However, the previous study (3) resuspended gametocytes in complete culture medium and incubated at 37°C for 2 h to allow recovery of microtubules. The authors should comment on whether the difference in protocol might be responsible for the different observations.

The gametocytes in our study were never allowed to cool down. A heat block was set up next to the plunger for vitrification of grids and cells were kept in this until being placed on the grids and immediately vitrified. We have clarified this in the methods section (page 32, line 660):

“Cells were kept warm in at 37°C heat block next to the plunge freezer.”

See above comment for our interpretation of discrepancy in measurements.

6) Page 20 Line 379. The authors state: “Thus, it is likely that there is a unique protein, linking the SPMTs to the IMC in all parasite forms analysed.” The data are not sufficient to make such a strong statement.

Thank you for pointing this out, we fully agree there is insufficient data for such a statement. Nevertheless, we would like to keep this as a postulate. It now reads (page 20, line 391):

“Thus, we show that, although each stage has a unique SPMT structure, the distance between the SPMTs and IMC remains consistent between stages. We therefore hypothesise that a unique protein links the SPMTs to the IMC in all parasite forms analysed.”

7) The authors provide a detailed discussion of the implications of the findings. An additional point that would be worth discussing is the recent finding that SPMTs are polyglutamated (2) (1), which may contribute to the stability of SPMTs in the absence of (the presumed) γ TuRC stabilisation.

Thanks for the suggestions to add to our discussion. We have now mentioned this in the discussion (page 25, line 490):

“Gametocyte SPMTs have been shown to be polyglutamated, which may play a role in their stability^{29,43}.”

The authors could also discuss a recent study that suggested that the first set of SPMTs in gametocytes may be nucleated from the cytoplasmic surface of the MTOC (2).

Thank you. We are still unsure how to interpret this data, in particular as we see microtubules next to each other in gametocytes with different polarity. Nucleation from a single MTOC doesn't seem to fit with this observation (maybe there are actually two MTOCs?). We have added this into the discussion (page 29, line 583).

“Recent data suggests that, at least initial SPMTs in gametocytes, are nucleated at a nuclear membrane-located centriolar-plaque⁴³. How different diameter microtubules and our observed random polarity of SPMTs can be achieved with this model will still need to be explored.”

Minor comments.

Figure 1C. The antibodies used for fluorescence labelling should be pointed out in the figures and figure legends.

The lines are all fluorescently tagged and details of these are in the methods section.

Line 275. “Although some canonical microtubules were present, these only accounted for 9% of the population, while the majority of singlet microtubules had 17-protofilaments (40%) (Fig. 3C)”. Fig. 3C should be Fig. 5C.

Thanks, changed.

“Nuclear microtubules were seen frequently in stage III/IV parasites, and a full spindle pole body was present in one tomogram (Fig. 5C).” Fig. 5C should be Fig. 5B.

Thanks! Changed.

There are several instances in which the order of presentation of the data does not match the order of presentation of the figures. This should be corrected (where possible).

We have reordered the supplemental figures to match their order in the text (and to accommodate the two new supplemental figures). We wanted to make the main figures easily comparable so that if readers choose to only look at the figures, they could see each lifecycle stage on its own figure and compare these to each other. We have endeavored to correct the order where possible, but some text will remain slightly out of order of the figure panels.

I. Bertiaux E, Balestra AC, Bournonville L, Louvel V, Maco B, et al. 2021. Expansion microscopy provides new insights into the cytoskeleton of malaria parasites including the conservation of a conoid. PLoS biology 19:e3001020

2. Dearnley MK, Yeoman JA, Hanssen E, Kenny S, Turnbull L, et al. 2012. Origin, composition, organization and function of the inner membrane complex of *Plasmodium falciparum* gametocytes. *J Cell Sci* 125:2053-63
3. Li J, Shami GJ, Cho E, Liu B, Hanssen E, et al. 2022. Repurposing the mitotic machinery to drive cellular elongation and chromatin reorganisation in *Plasmodium falciparum* gametocytes. *Nat Commun* 13:5054
4. Tilney LG, Porter KR. 1967. Studies on the microtubules in heliozoa: II. The effect of low temperature on these structures in the formation and maintenance of the axopodia. *The Journal of cell biology* 34:327

Reviewer comments, second round –

Reviewer #2 (Remarks to the Author):

The authors present an impressive study with the great significance of showing the use of FIB milling and cryogenic electron tomography to study microtubule architecture and composition in situ.

The authors have responded to the suggested comments in an acceptable manner.

Minor comments

Fig S2

For the genomic locus (S2a), what is the meaning of the *spm1* that is positioned between the 3' *dhfr* and the 3' UTR?

Is the second primer the authors used to verify the 3' integration for *TrxL1* missing?

I understand the authors' concern about using antibody labeling. Still, they suggest that the patterning of the tagged *TrxL1* and *SPM1* is similar to that of labeled tubulin presented in reference 28. Since it is impossible to determine the polarity of the sporozoites, using a microtubule probe (as used in reference 28) to show colocalization is more convincing.

Reviewer #3 (Remarks to the Author):

This revised manuscript by Ferreira and colleagues successfully addresses most of my queries.

In response to the question about the discrepancy between the observation of unusual large microtubules (27-37 nm) in this study and a previous study that reported the diameter of SPMTs in gametocytes as 27 ± 4 nm (3), the authors state: "...this is the first study to accurately measure structures (rather than individual microtubules) solved in vivo at native (vitrified, unfixed and unstained) conditions. Many different reasons (including shrinkage due to resin embedding, heavy metal staining and sectioning artifacts) could result in the discrepancy in the measurements."

Another previous study used cryoET to examine gametocyte SPMTs (DOI: 10.1242/jcs.099002). That study reported microtubule singlets of 25 nm in diameter and doublets of 42 nm.

The authors provide further assurance that the "Cells were kept warm in at 37°C heat block." Nonetheless, it remains possible that the Percoll purification protocol may induce changes in the gametocyte cytoskeleton, that can be reversed by a recovery incubation. Thus, difference in sample prep could account for the differences observed with previous studies.

Given that the current finding is so unusual; it is important that the authors at least comment in the text on the fact that the findings are different to previous studies, including a cryoET study.

Nature Communications: Response to reviewers round 2

Reviewer #2 (Remarks to the Author):

The authors present an impressive study with the great significance of showing the use of FIB milling and cryogenic electron tomography to study microtubule architecture and composition in situ.

The authors have responded to the suggested comments in an acceptable manner.

We thank the reviewer for these comments.

Minor comments

Fig S2

*For the genomic locus (S2a), what is the meaning of the *spm1* that is positioned between the 3' *dhfr* and the 3' UTR?*

Single-crossover recombination was used to endogenously tag proteins in *Plasmodium berghei*. To use this strategy, approximately 1 kilobase of the C-terminal sequence (for C-terminal tagging) is used for recombination. This 1kb homology region ends up being duplicated within the tagged locus. Only the first gene sequence, which is linked to GFP, is expressed under the endogenous promoter. The second sequence is not expressed as it lacks a promoter. As both SPM1 and TrxL1 have ORF with a length shorter than 1 kilobase, the full open-reading frame was used for single-cross over recombination.

The second fragment should have been labelled “mip” instead of “spm1”. This was corrected in the updated figure.

Is the second primer the authors used to verify the 3' integration for TrxL1 missing?

As 3' integration for TrxL1 and SPM1 was confirmed using primer P2239 (in combination with P650 for SPM1 and P2221 for TrxL1, respectively), primer P2239 was indicated only once. To prevent misinterpretation, primer P2239 is now indicated twice in the updated figure to show primer combinations for 3' integration for SPM1 and TrxL1 separately.

I understand the authors' concern about using antibody labeling. Still, they suggest that the patterning of the tagged TrxL1 and SPM1 is similar to that of labeled tubulin presented in reference 28. Since it is impossible to determine the polarity of the sporozoites, using a microtubule probe (as used in reference 28) to show colocalization is more convincing.

We agree that a colocalization with a microtubule probe would be required were these data to be presented in isolation. However, given that the fluorescence labelling is supporting evidence for our tomography and subtomogram averaging data intended to show the presence of TrxL1 in sporozoites, we believe that it is sufficient in its current form.

We agree, however, that the polarity of the sporozoite is ambiguous in our figure and so have removed the words “with localisations consistent with sporozoite SPMTs” (page 7, line 170) from the manuscript. We have also toned down the language, making it clear that the proteins that we suggest make up the luminal helix are from a hypothesis that is likely the case but not unambiguously confirmed.

Reviewer #3 (Remarks to the Author):

This revised manuscript by Ferreira and colleagues successfully addresses most of my queries.

We thank the reviewer for the comments.

In response to the question about the discrepancy between the observation of unusual large microtubules (27-37 nm) in this study and a previous study that reported the diameter of SPMTs in gametocytes as 27 ± 4 nm (3), the authors state: “..this is the first study to accurately measure structures (rather than individual microtubules) solved in vivo at native (vitrified, unfixed and unstained) conditions. Many different reasons (including shrinkage due to resin embedding, heavy metal staining and sectioning artifacts) could result in the discrepancy in the measurements.”

Another previous study used cryoET to examine gametocyte SPMTs (DOI: 10.1242/jcs.099002). That study reported microtubule singlets of 25 nm in diameter and doublets of 42 nm.

The diameters we reported are measured from subvolume averages ie: the mean of a large number of microtubules with the same number of protofilaments, but collected with varying defoci. Previous measurements were performed directly from tomograms. This is associated with errors as the structures cannot be directly interpreted as density due to signal delocalisation caused by the contrast transfer function.

However, we do agree that it is beneficial to mention that our results are in contrast to previous measurements and have added a sentence in the results section: “This was unexpected as in all organisms studied to date, cytoplasmic microtubules are composed of a single tube (singlet microtubule) and previous studies reported smaller diameter microtubules³¹.” (pg 10, line 271)

The authors provide further assurance that the "Cells were kept warm in at 37°C heat block." Nonetheless, it remains possible that the Percoll purification protocol may induce changes in the gametocyte cytoskeleton, that can be reversed by a recovery incubation. Thus, difference in sample prep could account for the differences observed with previous studies.

h

Given that the current finding is so unusual; it is important that the authors at least comment in the text on the fact that the findings are different to previous studies, including a cryoET study.

Thank you, we have added a sentence in the results, highlighting that our measurements are not in agreement with previous studies (see above comment) and have further clarified the methods section.

“Isolated gametocytes were washed twice and then resuspended in pre-warmed RPMI without albumax, serum or phenol red and kept warm (for between 15 minutes and 45 minutes) until immediately prior to freezing.